materials science

graphene oxide, long-term ageing, structural influence, thermal decomposition

**Author for correspondence:**
Shiguo Du
e-mail: d18031847852@163.com

# Effect of long-term ageing on graphene oxide: structure and thermal decomposition

Chen Li[1,2], Yanling Lu[1], Jun Yan[3], Weibo Yu[1], Ran Zhao[1], Shiguo Du[1] and Ke Niu[1]

[1]Army Engineering University of People's Liberation Army, Shijiazhuang Campus, No. 97 Heping West Road of Shijiazhuang, Shijiazhuang, Hebei province 050003, People's Republic of China
[2]Institute of Chemical Defence, Academy of Military Science, Zhijiang, Hubei province 443200, People's Republic of China
[3]Department of Road and Bridge Engineering, Transportation Vocational and Technical College, Shijiazhuang, Hebei province 050003, People's Republic of China

 CL, 0000-0002-0203-0882; SD, 0000-0002-6211-5431

After long-term ageing, the structure of graphene oxide prepared by the modified Hummers method changed. Because of the desorption of oxygen-containing functional groups, the C/O ratio of graphene oxide increased from 1.96 to 2.76. However, the average interlayer distance decreased from 0.660 to 0.567 nm. The content of -CH- and -CH$_2$- decreased; however, the type of oxygen-containing functional groups did not change. Moreover, $I_D/I_G$ increased from 0.87 to 0.92, indicating that the defect density decreased because of desorbing oxygen functional groups after ageing. When the temperature exceeded 60°C, CO$_2$ produced by decomposing graphene oxide was detected. The thermal decomposition changed after ageing. The decomposition peak temperature decreased from 216°C to 195°C. The CO$_2$ amount produced remained almost unchanged; however, the amount of CO, SO$_2$ and H$_2$O decreased. After ageing, the apparent activation energy of graphene oxide decreased from 150 to 134 kJ mol$^{-1}$.

## 1. Introduction

Graphene oxide, commonly referred to as functionalized graphene, is a graphene derivative with a six-membered ring structure similar to that of graphene. Currently, researchers believe that oxygen-containing groups of graphene oxide are primarily the hydroxyl, carboxyl and epoxy groups [1]. Furthermore, there may be carbonyl groups, semiquinones, tertiary carbon hydroxyl groups, ethers, quinones, furan-like bridges and five-membered lactones [2]. Generally, the epoxy and hydroxyl groups are located on the basal plane of graphene

oxide, the carboxyl groups are located on the edge of graphene oxide, and the hydroxyl groups also appear on the edge of graphene oxide [3,4]. The presence of these active groups makes it easy for graphene oxide to interact with other substances.

Graphene oxide is a high-energy, thermally unstable material prone to the disproportionation reaction even under mild heating conditions; it emits a considerable amount of heat during the thermal decomposition process [5]. Because of the energetic properties of graphene oxide, its use in preparing composite energetic materials can avoid a reduction in the reaction heat of these materials. Graphene oxide has a large specific surface area ($1300 \, \mathrm{m^2 \, g^{-1}}$) [6], which allows it to be closely combined with other components in the composite energetic material, thereby increasing the reaction rate of the material [3,7]. Furthermore, the active groups (hydroxyl, carboxyl and epoxy) in graphene oxide make it easy to prepare graphene oxide containing composite energetic materials. These advantages make graphene oxide extremely suitable for preparing composite energetic materials.

In recent years, there have been many studies using graphene oxide to prepare energetic materials and composite energetic materials [7–12]. Usually, energetic materials are stored as ammunition and pyrotechnics for years or even decades. However, at room temperature, graphene oxide has been demonstrated to be metastable and begins to decompose at a lower temperature (50°C) [13,14]. Furthermore, because graphene oxide has a large specific surface area, it can easily absorb moisture in air. This indicates that the properties of graphene oxide may change with an increase in storage time, which may affect the performance of graphene-based energetic composites. Therefore, it is important to examine the influence of ageing on the structure and thermal decomposition of graphene oxide.

Many studies have been conducted on the ageing properties of graphene oxide. Iakunkov *et al.* examined the swelling phenomenon of graphene oxide films prepared using the Hummers method after ageing in air, water and different alcohols [15]. Furthermore, Yin *et al.* reported that ageing does not significantly affect the decomposition enthalpy of graphene oxide, but slightly reduces the initial decomposition temperature [16]. Holt *et al.* [17] examined the ageing and degradation of graphene oxide in water, while Channei *et al.* [6] examined the effects of different oxidation ageing times (12, 24 and 72 h) on the structure and properties of graphene oxide. Zhou & Bongiorno [18] used first-principles probabilistic calculation methods to examine the low-temperature process of graphene oxide decomposition and the effect of ageing on the structure and chemical and dynamic stability of materials.

Although long-term ageing is a tedious method, it can better approximate the actual use of graphene oxide. However, studies on the effects of long-term ageing on the structure and thermal decomposition of graphene oxide have not been reported.

# 2. Material and methods

## 2.1. Materials

Natural flake graphite, potassium permanganate, concentrated sulfuric acid, phosphoric acid, 5% hydrochloric acid and barium chloride were used.

## 2.2. Preparation of graphene oxide

The preparation of graphene oxide can be roughly divided into the following three processes: intercalation, oxidation and exfoliation. Common preparation methods of graphene oxide are the Brodie method, Staudenmaier method, Hoffman method, Tour method and Hummers method [19]. Because of its advantages of safety and reaction efficiency, the Hummers method and various improved Hummers methods are currently the most widely used methods [20,21]. In this study, an improved Hummers method was used to prepare graphene oxide.

First, 108 ml of concentrated $H_2SO_4$ and 12 ml of $H_3PO_4$ were mixed and placed in a water bath at 25°C. Then, 5 g of flake graphite was added to the mixture and stirred for 30 min. Subsequently, 30 g of $KMnO_4$ was added slowly to the mixture and stirring continued for 12 h until the solution turned dark green. Then, 18 ml of $H_2O_2$ was slowly added to the mixture to remove excess $KMnO_4$, and the solution was stirred for 30 min until it cooled. After the mixture was suction filtered, a large amount of deionized water and HCl were added for washing. The suction filtration and washing step were repeated several times until the sulfate ions could not be detected after adding $BaCl_2$. Then, the filter cake was added to 250 ml of deionized water under ultrasonic stirring for 4 h. During the ultrasonic

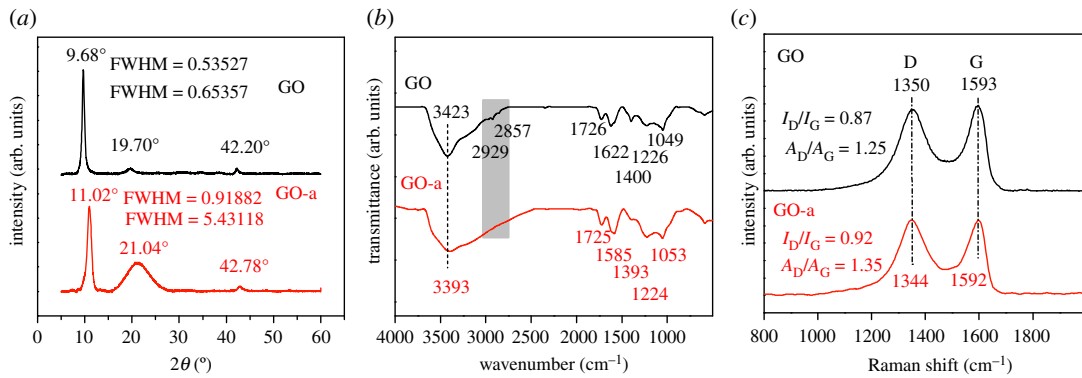

**Figure 1.** (*a*) XRD, (*b*) FTIR and (*c*) Raman curves of graphene oxide prepared after 5 days (GO; black) and graphene oxide aged for 2 years (GO-a; red).

dispersion process, biological ice packs are required to be continuously replaced to avoid excessive temperature. The beaker was taken out and freeze-dried for 24 h to obtain graphene oxide.

## 2.3. Preparation of long-term-aged graphene oxide

The graphene oxide prepared above was stored in a constant temperature and humidity box for 2 years and then removed for characterization. The humidity box was purchased from MS SHIMEI Electric (Jiangsu, China). The temperature of the constant temperature and humidity box was set to 25°C, and the humidity was maintained in the range of 20–70%.

In this study, GO refers to newly prepared graphene oxide, while GO-a refers to graphene oxide aged for 2 years.

## 2.4. Characterization

X-ray diffraction (XRD) was performed on a Rigaku Ultima IV X-ray diffractometer operated at 44 kV and 44 mA using copper X-ray radiation ($\lambda = 1.5438$ Å). Samples were scanned from 5° to 60° ($2\theta$) with a step size of 0.02° and scanning speed of 5° min$^{-1}$. The Fourier transform infrared (FTIR) spectrum was obtained using a Bruker TENSOR II FTIR spectrometer (scanning range: 4000–500 cm$^{-1}$, Bruker, Germany). The samples were prepared by the KBr tablet method, and the mass ratio of graphene oxide to KBr was 1 : 100. The defect density of graphene oxide was characterized using a HORIBA Jobin Yvon LabRAM HR Evolution laser confocal Raman spectrometer with a laser wavelength of 532 nm and a 50× objective lens. The layer thickness of graphene oxide was characterized using a Bruker NanoManVS type atomic force microscope (AFM). The graphene oxide aqueous solution with a concentration of 0.01 mg ml$^{-1}$ was ultrasonically dispersed for 20 min, and then dropped on a mica sheet and dried at room temperature. The probe was then operated in tapping mode. Elemental analysis was performed on an X-ray photoelectron spectrometer (XPS) with an Al Ka anode (PHI Quantera II, ULVAC–Phi, Japan). The components of the gaseous products were detected by STA 449F3 thermal analysis combined with an infrared and mass spectroscopy system (TG–FTIR–MS, Netzsch, Germany) at a heating rate of 20°C min$^{-1}$ from 50°C to 900°C in Ar. The flow rate of argon was 50 ml min$^{-1}$. The infrared spectrum scanning range was from 4000 to 650 cm$^{-1}$, and the gas pool temperature was 200°C. The heat of decomposition of graphene oxide was measured by a Netzsch STA 449 TG–DSC synchronous thermal analyser where the carrier gas was N$_2$ with a flow rate of 50 ml min$^{-1}$. The heating range was from room temperature to 320°C, and the heating rate was 5, 10, 15 and 20°C min$^{-1}$.

## 3. Results and discussion

After storing for 2 years at 25°C and a relative humidity ranging from 20 to 70%, the structure of graphene oxide has undergone various changes.

As shown in figure 1*a*, the GO had a diffraction peak of the (002) plane close to 9.68°, while GO-a had a diffraction peak of the (002) plane close to 11.02°. This result indicates that the adsorbed water and part

of the oxygen-containing groups desorbed during the ageing process, which reduced the layer spacing. After ageing, the full width at half maximum (FWHM) of the diffraction peak of the (002) plane of graphene oxide increased from 0.53527° to 0.91882°. The Debye–Scherrer formula was used to calculate the lattice size of graphene oxide. After ageing, the lattice size of graphene oxide decreased from 14.48 to 7.11 nm. Furthermore, GO had weak diffraction peaks close to 19.70° and 42.20°, corresponding to the (002) crystal planes of oxygen partly desorbed graphene oxide and the (100) crystal planes of graphite, respectively [22]. The diffraction peak of the (002) crystal plane in GO-a appeared close to 21.04°, and the peak intensity and width of the diffraction peak considerably increased.

Figure 1a shows the FWHM of peaks close to 20°. After ageing, the lattice size of partly reduced graphene oxide decreased from 12.16 to 1.47 nm. The increase in peak intensity close to 21.04° shows that additional oxygen-containing groups desorbed from GO-a. Graphene oxide heated at 200°C had a strong peak at approximately 20° [23]. After ageing, the interplanar spacing of highly oxidized graphene oxide decreased from 9.00 to 7.59 Å; however, the interplanar spacing of partially deoxygenated graphene oxide decreased from 4.51 to 4.23 Å (calculated by the Bragg equation), but it was still larger than the interplanar spacing of graphite, i.e. 3.36 Å [19].

The Debye–Scherer formula [24] is listed below:

$$L_a = \lambda 0.9 \beta^{-1} (\cos \theta)^{-1},  \qquad (3.1)$$

where $L_a$ is the average grain size perpendicular to the crystal plane, $\lambda$ is the X-ray wavelength, $\beta$ is the FWHM of the diffraction peak and $\theta$ is the diffraction angle.

The Bragg equation [25] is as below:

$$2d \sin \theta = n\lambda,  \qquad (3.2)$$

where $d$ is the interplanar spacing, $\theta$ is the incident angle, $n$ is the diffraction order and $\lambda$ is the X-ray wavelength.

Figure 1b shows the mid-infrared spectrum of graphene oxide. For the as-prepared graphene oxide, the infrared absorption peaks close to 3423 and 1400 cm$^{-1}$ are the stretching and bending vibration peaks of OH, respectively [26]. The infrared absorption peaks close to 1726, 1622, 1226 and 1049 cm$^{-1}$ are attributed to the stretching vibration peaks of the carbonyl group (C=O), the aromatic ring skeleton vibration peak, and the stretching vibration peaks of the epoxy and C–O bonds, respectively [27]. The infrared absorption peaks near 2857 and 2929 cm$^{-1}$ correspond to the C–H stretching vibration of the CH$_2$ group [26,28,29], and the infrared absorption peak close to 2977 cm$^{-1}$ corresponds to the antisymmetric stretching vibration of CH$_3$ [30]. Compared with the as-prepared graphene oxide, the infrared absorption peaks of CH$_3$ and CH$_2$ in the infrared spectrum of GO-a are not obvious. This result shows that CH$_3$ and CH$_2$ were degraded during the ageing process. In the infrared spectrum of GO-a, the absorption peaks at 3393, 1393, 1725, 1586, 1224 and 1053 cm$^{-1}$ correspond to the stretching vibration mode of OH, the bending vibration mode of OH, the stretching vibration mode of the carbonyl group (C=O), the stretching vibration mode of aromatic C=C bonds, the stretching vibration mode of epoxy and the C–O bond, respectively. These results show that the type of oxygen-containing functional groups in graphene oxide did not change after ageing. Because of the desorption of surface water, the infrared vibration peak caused by OH bending vibration and aromatic C=C bonds close to 1622 cm$^{-1}$ was red-shifted to 1585 cm$^{-1}$.

Figure 1c shows the Raman spectrum of graphene oxide. The D band (1349 cm$^{-1}$) of graphene oxide is the respiration mode of A$_{1g}$ symmetric k-point phonons, originating in the graphite region with a small domain size, which is believed to reflect the local defects and disorder of the crystal lattice, particularly the defects and disorder at the edges of the graphite sheet. The G band (1597 cm$^{-1}$) is usually attributed to the first-order E$_{2g}$ phonon plane vibration of sp$^2$ hybridized carbon atoms, which corresponds to the plane stretching of the graphite lattice [28]. Many researchers consider that the larger the $I_D/I_G$ ratio, the greater the defect density of graphene oxide; however, as per studies by Cancado et al. [31] and Lucchese et al. [32], this view is incorrect. After ageing, the $I_D/I_G$ value of graphene oxide increased from 0.87 to 0.92. When the average defect distance $L_D$ is less than 4 nm, materials with additional defects have a smaller value of $I_D/I_G$. During the ageing progress, the formation of CO$_2$ created vacancy defects on the basal plane of graphene oxide; however, the reduction in the number of oxygen-containing functional groups was considerably greater than the formation of vacancy defects, which makes the average defect distance ($L_D$) of GO-a; i.e. the defect density decreased.

Figure 2 shows AFM images of graphene oxide. Figure 2a,c shows the morphology of GO and GO-a, respectively. Graphene oxide had an irregular flake structure; after ageing, the size of graphene oxide

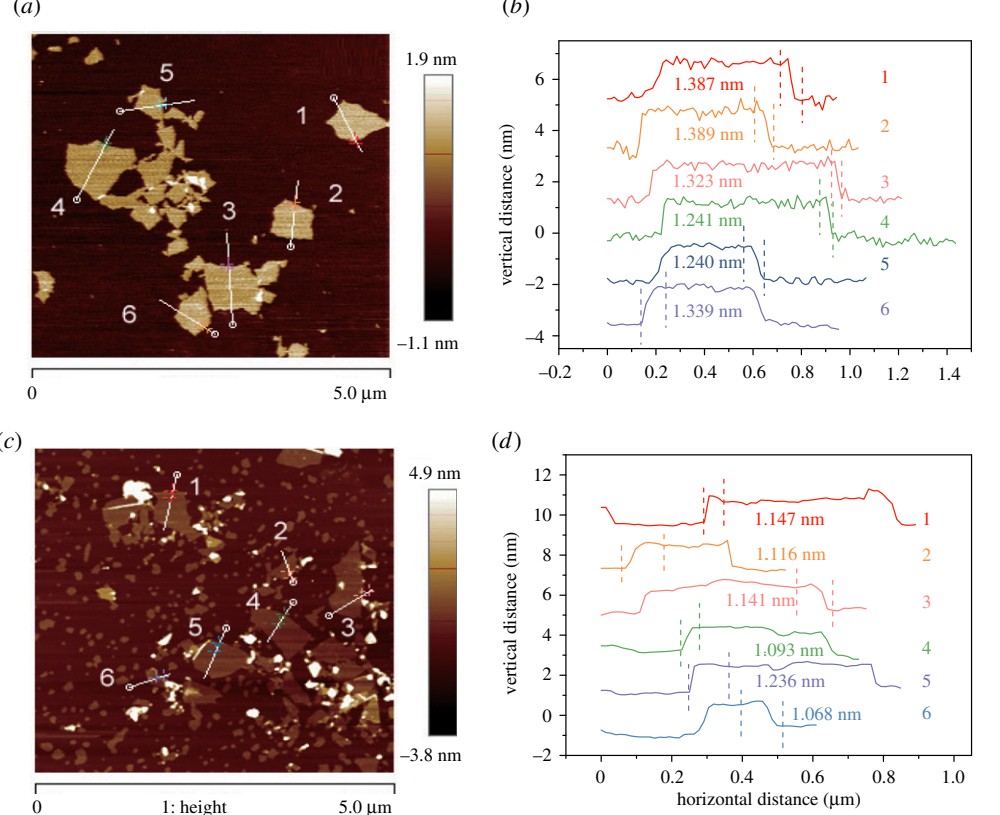

**Figure 2.** AFM images of graphene oxide prepared after 5 days (GO) and graphene oxide aged for 2 years (GO-a). The topography of graphene oxide: (*a*) GO and (*c*) GO-a. Interlayer spacing of graphene oxide: (*b*) GO and (*d*) GO-a.

decreased. The research results of Acik *et al*. demonstrated that the epoxy groups on graphene relax carbon bonds when they are arranged during the oxidation process. Ether groups are generated when the epoxy groups decompose, leading to the unzipping of graphene oxide [2]. Figure 2*b,d* shows the thickness curves of GO and GO-a, respectively. Previous studies reported that the thickness of single-layer graphene oxide ranges from 0.5 to 1.2 nm. Therefore, the average thickness of GO (1.320 nm) and GO-a (1.134 nm) in figure 2*b,d* may belong to two-layer graphene oxide; i.e. the thickness of single-layer GO was 0.660 nm, which decreased to 0.567 nm of GO-a after ageing. The decrease in sheet thickness can be attributed to the desorption of oxygen functional groups.

The surface elements of graphene oxide were characterized by a ULVAC PHI Quantera II XPS with an Al K$\alpha$ anode. The corrected binding energy of C 1s was 284.6 eV. Figure 3*a* shows the survey XPS spectra of graphene oxide. The intensity of the O 1s peak of GO-a was significantly lower than that of GO. After ageing, the C/O ratio of graphene oxide increased from 1.96 to 2.67; the results revealed that graphene oxide lost part of its oxygen-containing groups at 25°C and humidity of not greater than 70%. These results correspond to the XRD test.

Figure 3*b* shows a magnified view of the box in figure 3*a*. The peaks close to 231 and 168 eV are attributed to the 2s and 2p peaks of sulfur, respectively. Figure 3*c,d* present the C 1s and O 1s spectra of graphene oxide after deconvolution, respectively. After deconvolution of the C 1s spectrum of graphene oxide, there are five peaks close to 284.5 ± 0.1, 285.4 ± 0.2, 286.5 ± 0.3, 287.6 ± 0.2 and 288.9 ± 0.3 eV, corresponding to the sp$^2$ hybrid carbon atom (C=C, pink), sp$^3$ hybrid carbon atom (C–C, orange), hydroxyl/epoxy/ether (C–OH, C–O–C, R–O–R′, green), carbonyl/quinone (C=O, blue) and carboxyl group/ester/anhydride (O=C–OH, O=C–O, O=C–O–C=O, purple) [33], respectively. The O 1s spectrum of graphene oxide exhibits four peaks after deconvolution, corresponding to C=O (pink) close to 531.2 ± 0.3 eV, C–O (orange) close to 532.1 ± 0.2 eV and C–O–C close to 533.2 ± 0.3 eV (green) and adsorbed water close to 534.3 ± 0.4 eV (blue) [7]. The FWHM of each deconvoluted secondary peak is 1.45 eV.

After ageing, the C 1s spectrum of graphene oxide significantly changed and the peak area of the O 1s spectrum was significantly reduced. The content ratio of each group can be estimated as per the area of the secondary peak [34]. Table 1 presents the positions and relative areas of the graphene oxide C 1s and

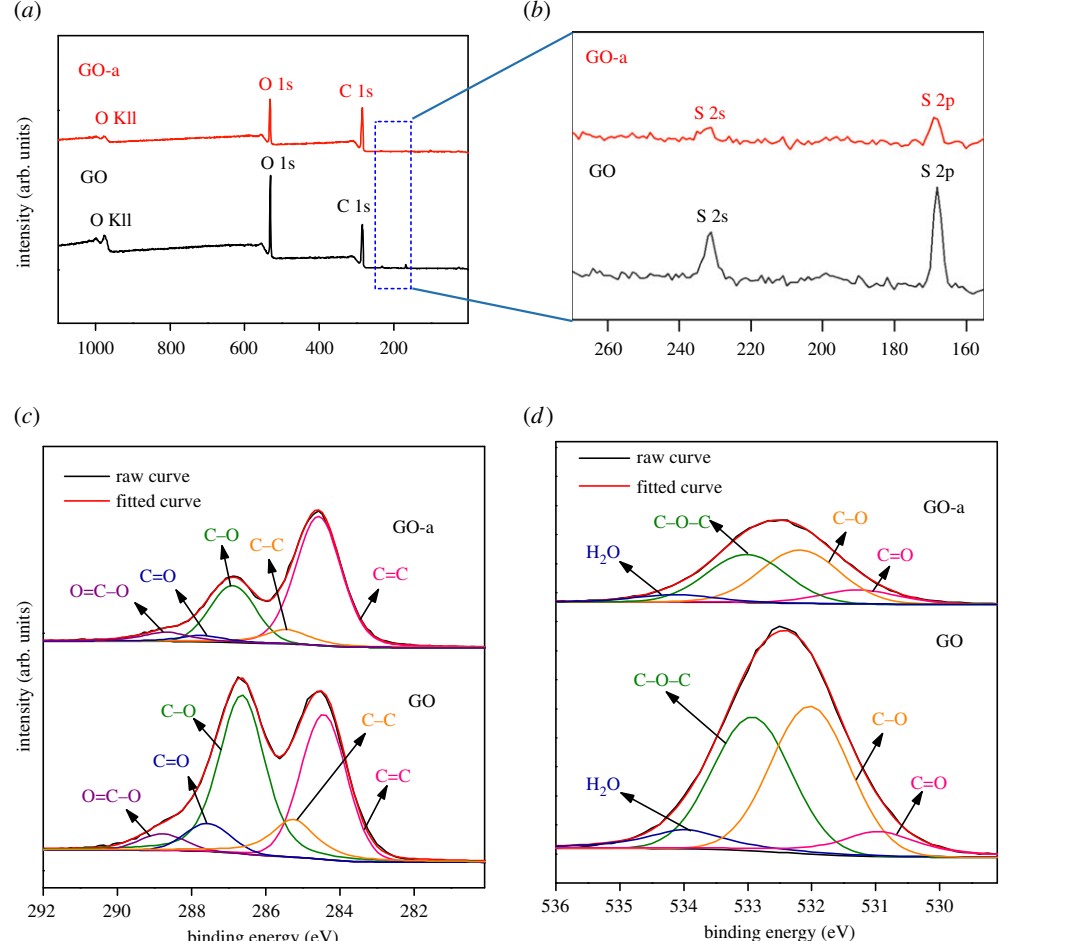

**Figure 3.** (a) Survey XPS spectra; (b) magnified graph of area inside box in (a); detailed (c) C 1s and (d) O 1s XPS spectra of graphene oxide.

O 1s deconvolution fitting peaks. From the relative area of the sub-peaks of the C 1s peak of GO-a in table 1, the relative content of $sp^2$ carbon atoms significantly increased, while the relative content of C–O significantly decreased. This indicates that a large amount of oxygen-containing functional groups is desorbed to increase the relative content of aromatic domains. This result agrees with that observed in the XRD and Raman spectra. The relative content of $sp^3$ carbon atoms decreased from 12.11 to 8.29%, indicating that the content of aliphatic carbocyclic structures decreased, which is consistent with the weakening of the corresponding $CH_2$ spectral intensity in the FTIR spectra. After ageing, the relative content of C=O decreased from 6.48 to 3.73%, and the relative content of O=C–O increased from 3.80 to 5.48%. The relative area of the sub-peak of the O 1s peak of GO-a and the absolute content of all oxygen-containing functional groups decreased.

Zhou & Bongiorno [18] demonstrated through first-principles calculations that the most effective low-temperature decomposition reaction of graphene oxide is the reaction between a pair of epoxides or hydroxyl groups chemically adsorbed on the same side of graphene. This may explain why the content of both C–O and C–O–C in graphene oxide decreases after ageing. Water is possibly involved in the degradation process of oxygen-containing functional groups [2,35]. The reaction of graphene oxide with water causes the C–C bond to break and transform into a humic acid-like structure [36]. Interestingly, C=O, which is considered to be more stable, has a similar decrease in C–O. A study by Acik *et al*. [35] demonstrated that epoxy moves to the defect hole and removes the carbonyl group by forming $CO_2$. Another study by Zhou & Bongiorno revealed that the reaction of C–H with epoxide and hydroxyl occurs easily. They argued that these reactions may be attributed to the metastable properties of multilayer extended graphene oxide. This conclusion agrees with the lack of a C–H absorption peak in the infrared spectrum of GO-a [37].

Thermogravimetric (TG) curve of graphene oxide was obtained at a heating rate of 20°C min$^{-1}$, as illustrated in figure 4a. The thermal decomposition of graphene oxide was a disproportionation

**Table 1.** Positions and areas of graphene oxide C 1 s and O 1 s deconvolution peaks.

| graphene oxide | feature | C 1s | | | | | O 1s | | | |
|---|---|---|---|---|---|---|---|---|---|---|
| | | C (sp²) | C (sp³) | C–O | C=O | O=C–O | C=O | C–O | C–O–C | H₂O |
| GO | peak (eV) | 284.5 | 285.3 | 286.6 | 287.6 | 288.8 | 531.0 | 532.0 | 532.9 | 534.0 |
| | area (%) | 34.64 | 12.11 | 42.97 | 6.48 | 3.80 | 7.31 | 38.30 | 39.03 | 15.36 |
| GO-a | peak (eV) | 284.6 | 285.4 | 286.9 | 287.7 | 288.7 | 531.3 | 532.2 | 533.0 | 534.1 |
| | area (%) | 58.67 | 8.29 | 23.83 | 3.73 | 5.48 | 6.51 | 38.11 | 43.34 | 12.05 |

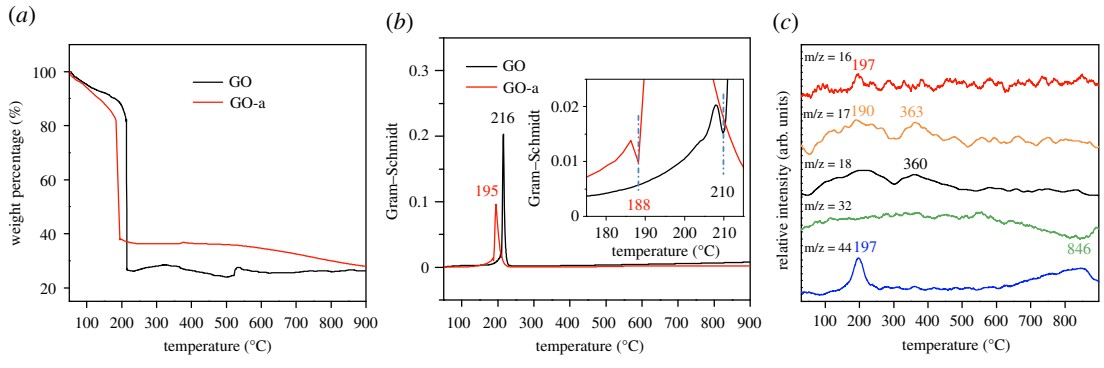

**Figure 4.** (a) TG curves of graphene oxide (GO and GO-a); (b) Gram–Schmidt gas absorbance during thermal decomposition; (c) mass spectrometry of gaseous products produced by GO-a ($m/z$ = 16, 17, 18, 32 and 44).

reaction. During the thermal decomposition process, a part of C atoms was oxidized to escape in the form of $CO_2$ or CO, while another part of C atoms was reduced. Calculating the second derivative of the TG curve to identify the corresponding inflection points of the curve could be used to roughly divide the decomposition stages of the sample. The thermal decomposition process of GO after preparation could be divided into four stages. The first stage was from 50°C to 112°C, the mass loss in this stage was approximately 5.92%, which was primarily attributed to the volatilization of adsorbed water. The second stage was from 112°C to 215°C. The weight loss in this stage was approximately 67.52%, which was mainly due to the desorption of oxygen-containing groups. The third stage was from 215°C to 522°C, and the weight in this stage first increased by approximately 1.93% (to approx. 323°C) and then decreased by approximately 3.72%. This may have been attributed to the adsorption of free radicals during the decomposition process and subsequent depth decomposition of intermediate products of graphene oxide. The fourth stage was from 522°C to 900°C. The weight in this stage first increased by approximately 2.76% (to 532°C) and then decreased by approximately 1.26%. This may reflect the process of adsorption and desorption of trace $O_2$ in the protective gas.

The thermal decomposition process of GO-a was similar to that of GO and could be roughly divided into four stages. The first stage was from 50°C to 147°C, and the weight loss was approximately 11.39%, primarily because of the volatilization of adsorbed water and the desorption of oxygen-containing functional groups. The second stage was from 147°C to 195°C, and the weight loss was approximately 50.82%. In this stage, the GO-a was violently decomposed to produce a large amount of gas, which blew away certain samples from the crucible. This occurred in the TG test of GO at a heating rate of 20°C min$^{-1}$. The third stage is from 195°C to 365°C, and the weight loss was approximately 1.50%. The fourth stage was from 365°C to 900°C, in which the weight first increased by 0.45% (to approx. 377°C) and then decreased by 8.74%, which may be related to the free decomposition process. The adsorption of radicals and the decomposition of carbon atoms caused by the reaction with trace oxygen in the carrier gas under high-temperature conditions were related. Zhang *et al*. [38] reported a similar phenomenon of graphene being oxidized at high temperature.

Figure 4b shows the relationship between gas generated during the thermal decomposition of graphene oxide and temperature. After ageing, the infrared signal of gas released by the thermal decomposition of graphene oxide was significantly weaker, indicating that the amount of gas produced was smaller. The peak temperature of gas generated by the decomposition of GO-a was close to 195°C; however, that of GO was close to 216°C. This demonstrates that ageing can reduce the thermal decomposition temperature of graphene oxide, which is consistent with the phenomenon reflected by the TG curve. Interestingly, before the violent decomposition of graphene oxide, the infrared absorption intensity of the gas produced by the decomposition of graphene oxide briefly decreased. As illustrated in the inset of figure 4b, for GO, this temperature point appears close to 210°C, while the corresponding temperature point for GO-a appears close to 188°C.

To examine the thermal decomposition process of GO and GO-a more comprehensively, contour plots were drawn of the infrared absorption signals of the decomposition products (figure 5a,b). Figure 5a shows that the gaseous decomposition products of graphene oxide exhibit a strong infrared absorption signal that corresponds to the severe weight loss during decomposition. As the temperature increases, the gaseous products produced by the decomposition of graphene oxide lead to a weak, continuous gaseous infrared absorption signal. This indicates that the thermal

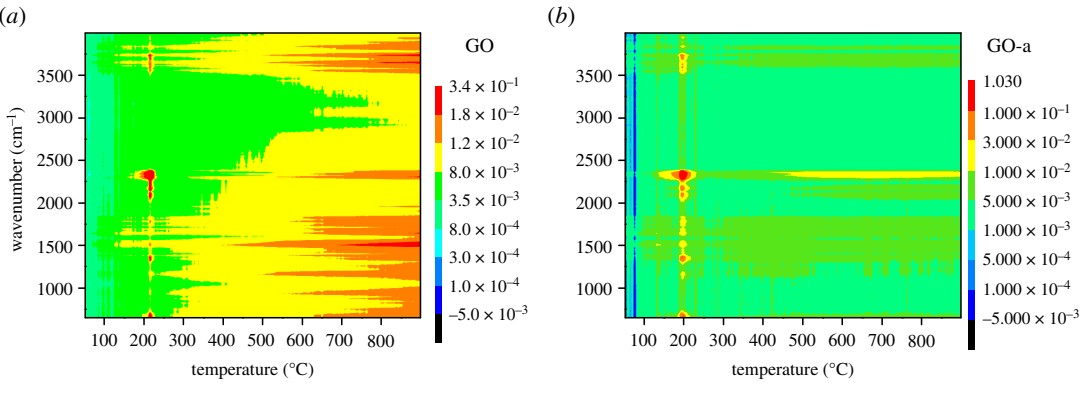

**Figure 5.** Contour plots of infrared absorption spectrum of gaseous products of (*a*) graphene oxide prepared after 5 days (GO) and (*b*) graphene oxide aged for 2 years (GO-a).

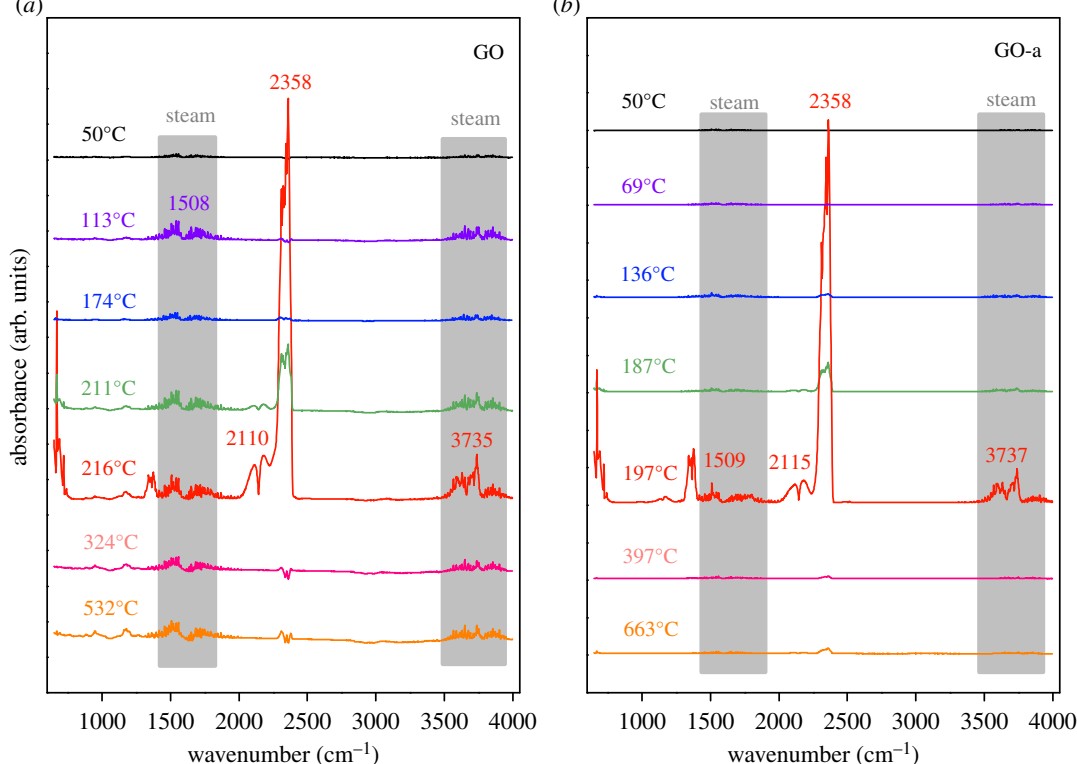

**Figure 6.** Infrared absorbance of gaseous products under different temperatures: (*a*) graphene oxide prepared after 5 days (GO) and (*b*) graphene oxide aged for 2 years (GO-a).

decomposition process of graphene oxide is a continuous and uninterrupted process when the temperature increases. The thermal decomposition process of GO-a illustrated in figure 5*b* is similar to that of GO. Both GO and GO-a are violently decomposed close to 200°C and maintain weaker decomposition under high-temperature conditions. The difference is that the temperature corresponding to the violent decomposition of GO-a is 195°C, whereas the temperature corresponding to that of GO is 216°C.

The three-dimensional infrared absorption spectrum of the gaseous decomposition products can be analysed by the temperature profile to examine the gas types at different temperatures, as illustrated in figure 6. In figure 6*a*, the decomposition products of GO at 216°C exhibit several obvious infrared absorption signals, which correspond to the infrared absorption of water vapour between 1400–1750 and 3483–3759 $cm^{-1}$; the infrared absorption peaks of $CO_2$ close to 671, 1377 and 2358 $cm^{-1}$; and the infrared absorption peaks of CO close to 2115 and 2183 $cm^{-1}$ [39–43]. Similarly, GO-a in figure 6*b* exhibits infrared absorption bands of $CO_2$, CO and water vapour at the corresponding positions

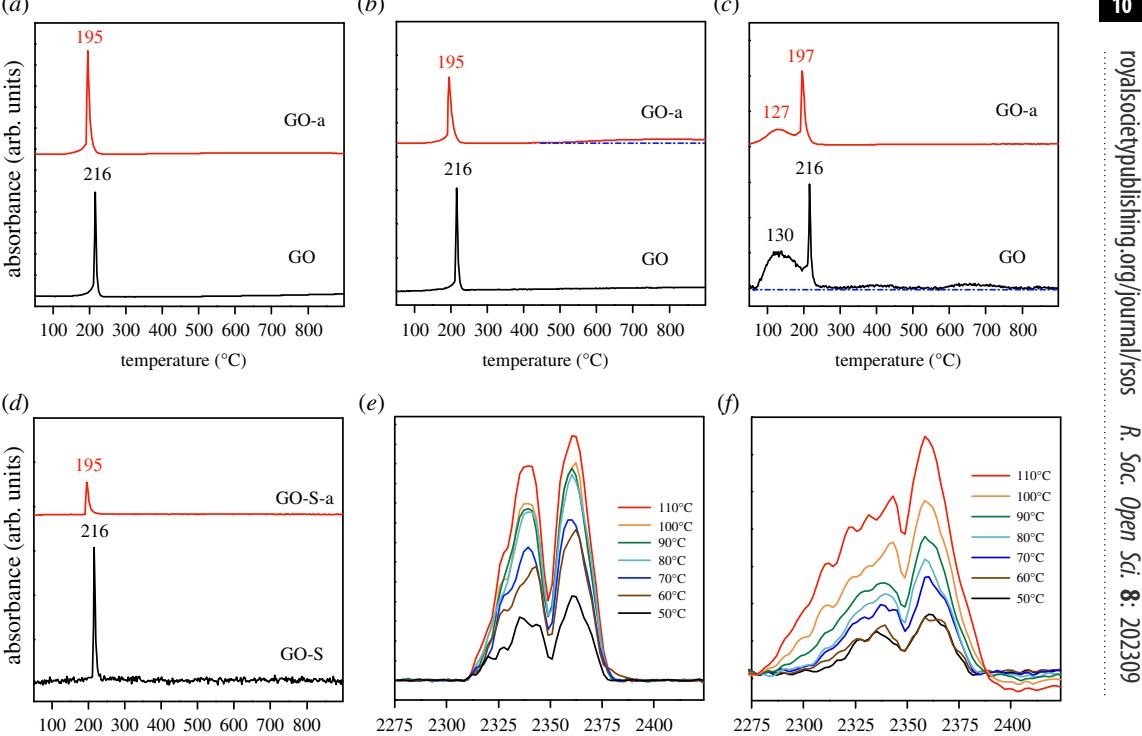

**Figure 7.** Infrared absorbance intensity of gaseous products under different temperatures: (a) $CO_2$, (b) CO, (c) steam and (d) $SO_2$. Infrared absorbance intensity of $CO_2$ under low temperature (less than 120°C): (e) graphene oxide prepared after 5 days (GO) and (f) graphene oxide aged for 2 years (GO-a).

during the thermal decomposition process at 195°C. Furthermore, it is difficult to completely remove sulfur during the preparation of graphene oxide. It is possible that the weak infrared absorption band close to 1168 cm$^{-1}$ in figure 6a,b corresponds to $SO_2$ [44]. The grey areas in figure 6a,b correspond to the infrared absorption signals of water vapour. The infrared absorption intensity of water vapour in figure 6a is obvious, while that in figure 6b is only obvious at the peak temperature (197°C). This indicates that the decomposition of GO produces more water vapour than the decomposition of GO-a.

To determine the relationship between the gaseous product content and temperature more accurately, infrared signals with greater intensity and less interference were selected to represent the corresponding gaseous products, such as $CO_2$ (2358 cm$^{-1}$), CO (2115 cm$^{-1}$), water vapour (1509 cm$^{-1}$) and $SO_2$ (1168 cm$^{-1}$). As per the Lambert–Beer law, infrared absorption intensity is positively correlated with substance content. Figure 7a–d presents the changes in infrared absorbance of $CO_2$, CO, water vapour and $SO_2$ with temperature, respectively. As illustrated in figure 7a–d, the $CO_2$, CO, water vapour and $SO_2$ generated by the decomposition of graphene oxide all have peaks during the violent decomposition, regardless of whether the sample has been aged. The difference is that the content of CO and water vapour generated by the decomposition of GO-a is slightly lower than that of GO, and the amount of $SO_2$ generated is considerably lower than that of GO; however, the amount of $CO_2$ generated does not appear to change.

In the middle-temperature section (greater than 300°C), almost no water vapour generated by the GO-a decomposition is observed. Compared with GO-a, GO contains more C-H and generates additional water vapour during the decomposition process. Furthermore, it decomposes to generate water vapour when the temperature is greater than 300°C. The decrease in CO content generated by the decomposition of GO-a may be related to the decrease in oxygen content. As illustrated in figure 7e, GO can be decomposed to produce $CO_2$ at 60°C, and the amount of production increases with an increase in temperature. In figure 7f, there is little difference between the infrared absorption intensity at 50°C and 60°C. The infrared intensity of $CO_2$ increases at 70°C, which indicates that GO-a may generate $CO_2$ close to 70°C. The corresponding temperature of GO is approximately 60°C.

To clarify the influence of ageing on kinetic parameters of the thermal decomposition of graphene oxide, non-isothermal (5, 10, 15 and 20°C min$^{-1}$) differential scanning calorimetry tests were

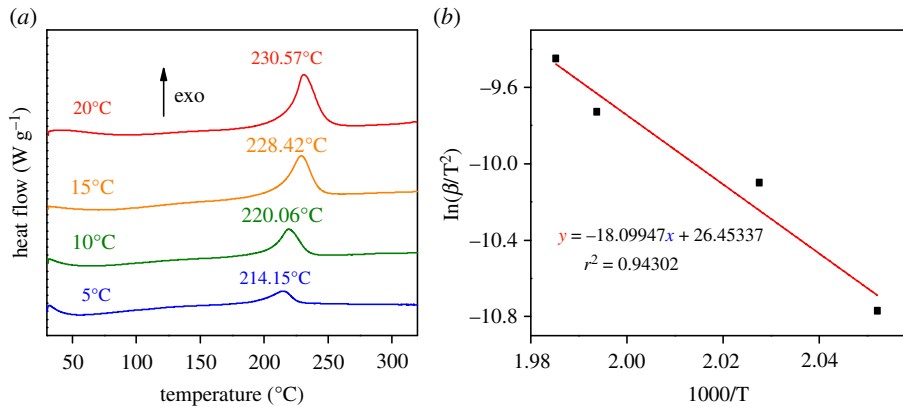

**Figure 8.** (*a*) Differential scanning calorimetry curves of graphene oxide prepared after 5 days (GO) and (*b*) fitting curve of In($\beta$/T$^2$) to 1000/T.


**Figure 9.** (*a*) Differential scanning calorimetry curves of graphene oxide aged 2 years (GO-a) and (*b*) fitting curve of In($\beta$/T$^2$) to 1000/T.

performed on GO and GO-a, as illustrated in figures 8 and 9, respectively. Their apparent activation energy ($E_a$) was calculated by the Kissinger formula. The results demonstrate that after ageing, the apparent activation energy ($E_a$) of graphene oxide decreased from 150 to 134 kJ mol$^{-1}$.

## 4. Conclusion

The results presented in this study suggest that the aliphatic hydrocarbon groups and oxygen-containing functional groups in the graphene oxide structure are degraded during long-term ageing. The average interlayer spacing of graphene oxide decreased from 0.660 to 0.567 nm. The process of desorption of physically adsorbed water in the graphene oxide interlayer during the ageing process caused the average layer spacing of graphene oxide to decrease. The desorption of water and oxygen-containing functional groups resulted in the appearance of graphene sheets with a lower degree of oxidation, which appeared as a broad peak at 21.04° in the XRD pattern. During ageing, the types of graphene oxide functional groups did not significantly change, but the proportion changed. The content of oxygen-containing functional groups significantly decreased. By comparing the amount of $CO_2$ produced at different temperatures, it could be inferred that graphene oxide may have begun to decompose at approximately 70°C. When the temperature exceeded 100°C, the weight loss rate of GO-a was significantly accelerated.

The results of thermal analysis–mass spectrometry indicated that the ion current intensity peak of OH radicals appeared at approximately 190°C during the decomposition of graphene oxide, and it then began to violently decompose at approximately 195°C. The lower decomposition temperature of graphene oxide limits its application under higher temperature conditions. The graphene oxide underwent slow structural changes when stored at room temperature, and the structural changes led

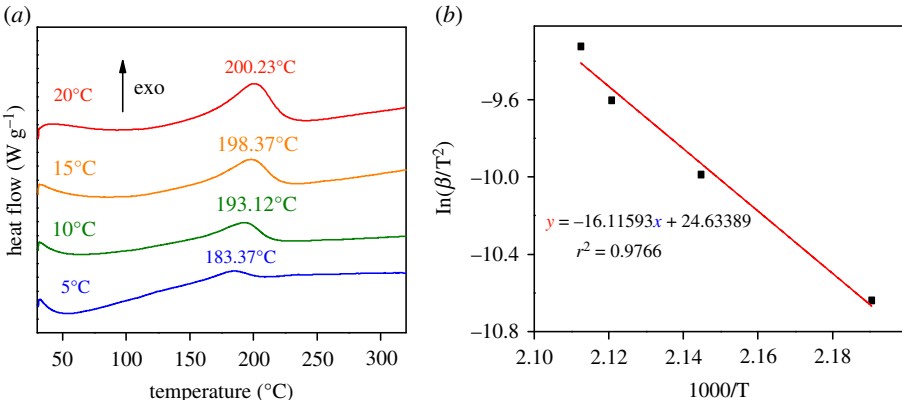

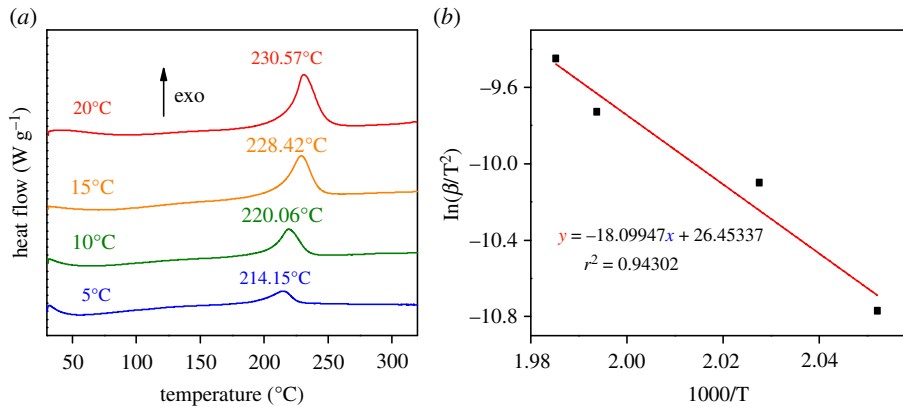

_

to changes in the physical and chemical properties. Prolonged contact with moisture in the air caused the activation energy of graphene oxide to decrease from 150 to 134 kJ mol$^{-1}$. Thus, after reaching a certain time limit, graphene oxide may not be able to meet the requirements of users. Because light affects the decomposition of graphene oxide [34], graphene oxide must be used in dark, dry and low-temperature conditions to ensure product stability during long-term use and storage.

Data accessibility. Data in this paper can be got at Dryad: https://doi.org/10.5061/dryad.ghx3ffbn0 [45].

Authors' contributions. C.L. was involved in writing, experiment, drawing and data analysis; Y.L. was involved in funding; J.Y. was involved in writing correction; W.Y. was involved in data processing; R.Z. was involved in data processing; S.D. was involved in experimental design and funding; K.N. was involved in writing correction.

Competing interests. We have no competing interests.

Funding. All the tests are funded by no. 51272284 National Natural Science Foundation of China.

Acknowledgements. The author would like to thank the support in testing technology from Tsinghua University and shiyanjia lab (www.shiyanjia.com).

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
