## [Peer Review File · Royal Society Open Science]

Review History

RSOS-202309.R0 (Original submission)

Review form: Reviewer 1

Is the manuscript scientifically sound in its present form?

Yes

Are the interpretations and conclusions justified by the results?

Yes

Is the language acceptable?

Yes

Do you have any ethical concerns with this paper?

Yes

Have you any concerns about statistical analyses in this paper?

No

Recommendation?

Accept with minor revision (please list in comments)

Comments to the Author(s)

1

Please further describe how to control or regulate defects and functional groups and composition during aging.

2

The theoretical analysis should be strengthened, and the change of adsorption energy and binding energy should be further further combined with the thermal analysis experiment or theoretical study in the aging process, so as to support the evolution behavior and related mechanism analysis of defects and functional groups.

3

According to the experimental results, including the peer experiments, the evolution mechanism of the composition structure in the aging process was further improved.

4

The comparative analysis of similar research work is very necessary, the author please make some supplement.

5

It is suggested that the highlights and significance of this paper should be further clarified, and the evolution law of defects, functional groups and composition structure during the aging process of GO should be emphasized, and the evolution mechanism should be revealed in combination with the ten revision previews of supporting literature. Related literature, *SMALL*, 2018, 14(29), 1800987; *ADV. MATER.*, 2014, 26(21): 3484-3489.

Review form: Reviewer 2

Is the manuscript scientifically sound in its present form?

Yes

Are the interpretations and conclusions justified by the results?

No

Is the language acceptable?

Yes

Do you have any ethical concerns with this paper?

No

Have you any concerns about statistical analyses in this paper?

Yes

Recommendation?

Major revision is needed (please make suggestions in comments)

Comments to the Author(s)

The authors report a detailed study on the changes in thermal decomposition and chemical structure of long-term aged GO under room condition. Particularly, the author studies extensively the thermal decomposition products and thermal stability of the aged GO, both of which provide useful information for academic and industrial users/producers of GO. Therefore, it is suitable for publication in RSOS after addressing the following comments:

1. When analyzing the FTIR result on page 4, the author stated that the absorption band at 1620 cm^{-1} is caused by the aromatic ring vibration. In most GO literature, the scissor bending mode of water can also contribute to the absorption at this wavenumber for GO. When moisture left the GO, this band downshifts to a lower wavenumber, which is what was observed for the 2-year aged GO.
2. The authors also observed -CH- stretching for pristine GO in the FTIR spectrum and uses the existence of -CH- stretching to explain the change in layer thickness post aging. The existence of -CH- is quite uncommon in literature. How can the authors be sure that it is not due to contamination? Did the authors collect signals from multiple different samples to confirm this observation?
3. Page 4, line 30 to 33: "After 2 years of aging, the halfwidth of the diffraction peak of the (001) plane of graphene oxide increased from 0.53527° to 0.91882° , which indicates that the size of the (001) plane of graphene oxide became smaller.". It is better to use interlayer spacing rather than size for more precise expression.
4. On Page 4, when reporting Raman results and calculating D/G ratio of the two GOs, the author concludes that the defect density of sp^2 region of GO increased based on an increasing D/G ratio from 0.85 to 1.03 of GO after 2 years of aging. Were there enough measurements carry out to support the increase in D/G ratio and is such increase significantly different statistically? The authors should also consider the correlation of D/G ratio vs. the mean inter-defect distance (LD) in the full LD range. (Canc -ado, L. G.; Jorio, A.; Ferreira, E. H. M.; Stavale, F.; Achete, C. A.; Capaz, R. B.; Moutinho, M. V. O.; Lombardo, A.; Kulmala, T.; Ferrari, A. C. Quantifying Defects in Gra- phene via Raman Spectroscopy at Different Excitation Energies. *Nano Lett.* 2011, 11, 3190-3196.) When LD drops below 3 nm, the D/G ratio starts to increase, and this could be applied to GO where the graphene basal plane is highly functionalized with different oxygen groups. In short, it is important to understand that an increase in D/G ratio is not always correlate with a decrease in LD.
5. On Page 5, the authors report XPS area calculation and areal ratio between pristine and aged GO for the deconvolution of C1s. This is not recommended as XPS area can vary from point to point even for the same sample. Instead, it is recommended that the authors compare the relative composition of each deconvoluted component between the two GOs. To be able to make such comparison, the authors need to deconvolute at least 3 high resolution C1s spectra for each GO and report standard deviations for each component.
6. Fig 4 can be better presented for better and more effective data visualization. The authors can consider plotting the TGA and MS data (combine Fig 4a and c) in one graph for better presentation of data. The authors can have two separate TGA-MS plots for the two different GOs. Please check Eigler, Siegfried, et al. "Formation and decomposition of CO_2 intercalated graphene oxide." *Chemistry of Materials* 24.7 (2012): 1276-1282. for reference. It will be easier for readers to understand which gas product is generated at different stages of the thermal decomposition if the data is presented in this manner.

7. The data set of the normalized ion current intensity vs. temperature curve of O radical ($m/z = 16$) is too noisy for any meaningful interpretation. Did the authors record data for water ($m/z = 18$)? Does the data of water give any useful information?
8. The data visualization of Fig 5 is poor. It is very difficult to read the X-Y axis and I would recommend plotting the data on a contour plot for better presentation.
9. Page 6, line 53 to 57: It is not clear what the authors meant by "weak and continuous signal" means and also the significance of a continuous and uninterrupted process. Please revise and clarify the above sentences.
10. The authors did extensive curve fitting of the fingerprint regime of the FTIR spectra with 17 components. As it stands, this looks overinterpreted. There is no agreed physical model for fitting for the fingerprint regime of GO in the literature and thus the fitting is highly subjective and could easily lead to misinterpretation of the data. If the authors would like to keep this part of the study, I suggest putting them in the supporting information and clearly explain the steps for baseline subtraction and peak fitting.
11. I suggest author also run DSC measurements of the different GOs and include this data in the study. The DSC measurement of aged GO will give useful information about the change in thermochemistry and kinetics of graphite oxide exothermic decomposition over long term aging, which is important for safety in large-scale storage and processing of GO. Please see Qiu, Yang, et al. "Thermochemistry and kinetics of graphite oxide exothermic decomposition for safety in large-scale storage and processing." Carbon 96 (2016): 20-28 for more details.
12. In general, the graphics of the figures are poor, please improve the quality.

Review form: Reviewer 3

Is the manuscript scientifically sound in its present form?

No

Are the interpretations and conclusions justified by the results?

No

Is the language acceptable?

Yes

Do you have any ethical concerns with this paper?

No

Have you any concerns about statistical analyses in this paper?

No

Recommendation?

Reject

Comments to the Author(s)

The authors report experimental results regarding long-term ageing and thermal decomposition of graphene oxide.

Due to the following concerns the manuscript cannot be accepted for publication in its present form.

The authors must present in more detail the state of the art related to ageing and decomposition of graphene oxide. Graphene oxide after synthesis is known to reach a quasi-equilibrium state with a constant C/O ratio and a stable structure. Reduction mechanisms at room temperature were reported to be highly inefficient, decomposition processes being endothermic. This manuscript report results which stand in contradiction with these features, stating that reduction of graphene oxide takes place during two years of storage at ambient temperature and 70% humidity. More detailed investigations are necessary to prove these conclusions. The presented results are limited to only two samples, one recently synthesized and another sample which was stored for two years. A much more accurate study would contain information related to samples synthesized simultaneously and their structures and composition investigated at regular time intervals.

Review form: Reviewer 4

Is the manuscript scientifically sound in its present form?

No

Are the interpretations and conclusions justified by the results?

No

Is the language acceptable?

No

Do you have any ethical concerns with this paper?

No

Have you any concerns about statistical analyses in this paper?

No

Recommendation?

Major revision is needed (please make suggestions in comments)

Comments to the Author(s)

Overall, the introduction is clear, but the result and discussion is too long and sometimes difficult to be comprehend. I recommend the authors to simplify the discussions and revise the structures of the sentences. The authors can also consider rearranging some of the minor data into supporting information. Besides that, the authors can use symbols for the samples. For example, "long-term aged GO" and "newly prepared graphene oxide" are too tedious to read.

Is the GO sample before aging characterized 2 years ago? Or a newly synthesized GO sample was characterized to be compared with the aged samples?

1. Can the authors provide a more accurate humidity range for example "xx to 70RH%", rather than just mentioning "not exceed 70%". This is because the humidity is very important for experimental repeatability.
2. Is the "humidity box" a commercial dry cabinet? Or is it a self-built device? It is important to provide a clear experimental description.
3. In Section 2.3, please briefly discuss the purpose of making the GO samples stirred in IPA and H₂O₂. Again, it is better to denote the samples with symbols.
4. In figure 1, which samples are compared? Is it a fresh GO or a 5 days aged GO? It is different with the other analysis?
5. In the XRD analysis, the authors stated that "After 2 years of aging, the halfwidth of the diffraction peak of the (001) plane of graphene oxide increased from 0.53527° to 0.91882°, which indicates that the size of the (001) plane of graphene oxide became smaller." What is "halfwidth"? Is it referring to the FWHM? Please use accurate term in the discussion. Moreover, how do the authors deduce that the size of GO became smaller? Is it proven by the results or referred to another study?
6. In the XRD results, it was stated that "The appearance of a broad peak indicates that there are some graphene flakes with lower oxidation degree in the graphene oxide sample aged for 5 years, which is similar to the XRD test results of the reduced graphite oxide at 200°C". This sentence is very confusing, please be clear when you are citing a literature results.
7. How is the interplanar spacing calculated in the XRD? It should be mentioned in the experimental section.
8. It was stated that "The result shows that after 2 years of aging, the content of graphitized carbon atoms in graphene oxide increases, and the order of arrangement along its stacking direction is poor." Why is the "order of arrangement" suddenly mentioned here without proper discussion earlier in the XRD results?
9. How is the C/O ratio in XPS determine? Was the relative sensitive factor incorporated in this calculatin?
10. What is the "area" data in Table 1 (XPS)? Is it the absolute area of the deconvoluted peaks? The absolute values of areas of two different samples should not be compared directly. The authors should instead compare the area% of the samples. For example, compare the area% of C=C in XPS C1s for "GO" and "aged GO".
11. In XPS O 1s, the amount of adsorbed water is discussed. However, XPS is operated under vacuum, therefore it is not a good analysis method for adsorbed water content.
12. There are many poorly written sentences in the discussion, for example "The thermal decomposition of graphene oxide is a disproportionation reaction, in which part of the carbon atoms is oxidized and the other part of the carbon atoms is reduced."
13. XPS shows that "aged GO" has adsorbed water, however TGA shows that "aged GO" has more % of adsorbed water?

14. The image quality of Fig5 a and b are very bad.

15. It was stated that "In addition, it is difficult to completely remove the impurity S element during the preparation of graphene oxide. We believe that the weak infrared absorption band near 1168 cm⁻¹ in Figure 6 (a) and Figure 6 (b) should correspond to SO₂" Was the sulfur element detected in XPS?

Decision letter (RSOS-202309.R0)

Dear Dr Li:

Title: The effect of long-term aging on graphene oxide: structure and thermal decomposition
Manuscript ID: RSOS-202309

The editor assigned to your manuscript has now received comments from reviewers. We would like you to revise your paper in accordance with the referee and Subject Editor suggestions which can be found below (not including confidential reports to the Editor). Please note this decision does not guarantee eventual acceptance.

Please submit your revised paper before 13-May-2021. Please note that the revision deadline will expire at 00.00am on this date. If we do not hear from you within this time then it will be assumed that the paper has been withdrawn. In exceptional circumstances, extensions may be possible if agreed with the Editorial Office in advance. We do not allow multiple rounds of revision so we urge you to make every effort to fully address all of the comments at this stage. If deemed necessary by the Editors, your manuscript will be sent back to one or more of the original reviewers for assessment. If the original reviewers are not available we may invite new reviewers.

On behalf of the Subject Editor Professor Anthony Stace and the Associate Editor Professor Tobias Hertel.

RSC Associate Editor:
Comments to the Author:
(There are no comments.)

RSC Subject Editor:
Comments to the Author:
(There are no comments.)

Reviewers' Comments to Author:

Reviewer: 1

Comments to the Author(s)

1

Please further describe how to control or regulate defects and functional groups and composition during aging.

2

The theoretical analysis should be strengthened, and the change of adsorption energy and binding energy should be further further combined with the thermal analysis experiment or theoretical study in the aging process, so as to support the evolution behavior and related mechanism analysis of defects and functional groups.

3

According to the experimental results, including the peer experiments, the evolution mechanism of the composition structure in the aging process was further improved.

4

The comparative analysis of similar research work is very necessary, the author please make some supplement.

5

It is suggested that the highlights and significance of this paper should be further clarified, and the evolution law of defects, functional groups and composition structure during the aging process of GO should be emphasized, and the evolution mechanism should be revealed in combination with the ten revision previews of supporting literature. Related literature, *SMALL*, 2018, 14(29), 1800987; *ADV. MATER.*, 2014, 26(21): 3484-3489.

Reviewer: 2

Comments to the Author(s)

The authors report a detailed study on the changes in thermal decomposition and chemical structure of long-term aged GO under room condition. Particularly, the author studies extensively the thermal decomposition products and thermal stability of the aged GO, both of which provide useful information for academic and industrial users/producers of GO. Therefore, it is suitable for publication in RSOS after addressing the following comments:

1. When analyzing the FTIR result on page 4, the author stated that the absorption band at 1620 cm^{-1} is caused by the aromatic ring vibration. In most GO literature, the scissor bending mode of water can also contribute to the absorption at this wavenumber for GO. When moisture left the GO, this band downshifts to a lower wavenumber, which is what was observed for the 2-year aged GO.

2. The authors also observed -CH- stretching for pristine GO in the FTIR spectrum and uses the existence of -CH- stretching to explain the change in layer thickness post aging. The existence of -CH- is quite uncommon in literature. How can the authors be sure that it is not due to contamination? Did the authors collect signals from multiple different samples to confirm this observation?

3. Page 4, line 30 to 33: "After 2 years of aging, the halfwidth of the diffraction peak of the (001) plane of graphene oxide increased from 0.53527° to 0.91882° , which indicates that the size of the (001) plane of graphene oxide became smaller.". It is better to use interlayer spacing rather than size for more precise expression.

4. On Page 4, when reporting Raman results and calculating D/G ratio of the two GOs, the author concludes that the defect density of sp^2 region of GO increased based on an increasing D/G ratio from 0.85 to 1.03 of GO after 2 years of aging. Were there enough measurements carry out to support the increase in D/G ratio and is such increase significantly different statistically? The authors should also consider the correlation of D/G ratio vs. the mean inter-defect distance (LD) in the full LD range. (Cancado, L. G.; Jorio, A.; Ferreira, E. H. M.; Stavale, F.; Achete, C. A.; Capaz, R. B.; Moutinho, M. V. O.; Lombardo, A.; Kulmala, T.; Ferrari, A. C. Quantifying Defects in Graphene via Raman Spectroscopy at Different Excitation Energies. *Nano Lett.* 2011, 11, 3190–3196.) When LD drops below 3 nm, the D/G ratio starts to increase, and this could be applied to GO where the graphene basal plane is highly functionalized with different oxygen groups. In short, it is important to understand that an increase in D/G ratio is not always correlate with a decrease in LD.

5. On Page 5, the authors report XPS area calculation and areal ratio between pristine and aged GO for the deconvolution of C1s. This is not recommended as XPS area can vary from point to point even for the same sample. Instead, it is recommended that the authors compare the relative composition of each deconvoluted component between the two GOs. To be able to make such comparison, the authors need to deconvolute at least 3 high resolution C1s spectra for each GO and report standard deviations for each component.

6. Fig 4 can be better presented for better and more effective data visualization. The authors can consider plotting the TGA and MS data (combine Fig 4a and c) in one graph for better presentation of data. The authors can have two separate TGA-MS plots for the two different GOs. Please check Eigler, Siegfried, et al. "Formation and decomposition of CO_2 intercalated graphene oxide." *Chemistry of Materials* 24.7 (2012): 1276-1282. for reference. It will be easier for readers to understand which gas product is generated at different stages of the thermal decomposition if the data is presented in this manner.

7. The data set of the normalized ion current intensity vs. temperature curve of O radical ($m/z = 16$) is too noisy for any meaningful interpretation. Did the authors record data for water ($m/z = 18$)? Does the data of water give any useful information?

8. The data visualization of Fig 5 is poor. It is very difficult to read the X-Y axis and I would recommend plotting the data on a contour plot for better presentation.

9. Page 6, line 53 to 57: It is not clear what the authors meant by "weak and continuous signal" means and also the significance of a continuous and uninterrupted process. Please revise and clarify the above sentences.

10. The authors did extensive curve fitting of the fingerprint regime of the FTIR spectra with 17 components. As it stands, this looks overinterpreted. There is no agreed physical model for fitting for the fingerprint regime of GO in the literature and thus the fitting is highly subjective and could easily lead to misinterpretation of the data. If the authors would like to keep this part of the study, I suggest putting them in the supporting information and clearly explain the steps for baseline subtraction and peak fitting.

11. I suggest author also run DSC measurements of the different GOs and include this data in the study. The DSC measurement of aged GO will give useful information about the change in thermochemistry and kinetics of graphite oxide exothermic decomposition over long term aging, which is important for safety in large-scale storage and processing of GO. Please see Qiu, Yang, et al. "Thermochemistry and kinetics of graphite oxide exothermic decomposition for safety in large-scale storage and processing." Carbon 96 (2016): 20-28 for more details.

12. In general, the graphics of the figures are poor, please improve the quality.

Reviewer: 3

Comments to the Author(s)

The authors report experimental results regarding long-term ageing and thermal decomposition of graphene oxide.

Due to the following concerns the manuscript cannot be accepted for publication in its present form.

The authors must present in more detail the state of the art related to ageing and decomposition of graphene oxide. Graphene oxide after synthesis is known to reach a quasi-equilibrium state with a constant C/O ratio and a stable structure. Reduction mechanisms at room temperature were reported to be highly inefficient, decomposition processes being endothermic. This manuscript report results which stand in contradiction with these features, stating that reduction of graphene oxide takes place during two years of storage at ambient temperature and 70% humidity. More detailed investigations are necessary to prove these conclusions. The presented results are limited to only two samples, one recently synthesized and another sample which was stored for two years. A much more accurate study would contain information related to samples synthesized simultaneously and their structures and composition investigated at regular time intervals.

Reviewer: 4

Comments to the Author(s)

Overall, the introduction is clear, but the result and discussion is too long and sometimes difficult to be comprehend. I recommend the authors to simplify the discussions and revise the structures of the

sentences. The authors can also consider rearranging some of the minor data into supporting information. Besides that, the authors can use symbols for the samples. For example, "long-term aged GO" and "newly prepared graphene oxide" are too tedious to read.

Is the GO sample before aging characterized 2 years ago? Or a newly synthesized GO sample was characterized to be compared with the aged samples?

1. Can the authors provide a more accurate humidity range for example "xx to 70RH%", rather than just mentioning "not exceed 70%". This is because the humidity is very important for experimental repeatability.
2. Is the "humidity box" a commercial dry cabinet? Or is it a self-built device? It is important to provide a clear experimental description.
3. In Section 2.3, please briefly discuss the purpose of making the GO samples stirred in IPA and H₂O₂. Again, it is better to denote the samples with symbols.
4. In figure 1, which samples are compared? Is it a fresh GO or a 5 days aged GO? It is different with the other analysis?
5. In the XRD analysis, the authors stated that "After 2 years of aging, the halfwidth of the diffraction peak of the (001) plane of graphene oxide increased from 0.53527° to 0.91882°, which indicates that the size of the (001) plane of graphene oxide became smaller." What is "halfwidth"? Is it referring to the FWHM? Please use accurate term in the discussion. Moreover, how do the authors deduce that the size of GO became smaller? Is it proven by the results or referred to another study?
6. In the XRD results, it was stated that "The appearance of a broad peak indicates that there are some graphene flakes with lower oxidation degree in the graphene oxide sample aged for 5 years, which is similar to the XRD test results of the reduced graphite oxide at 200°C". This sentence is very confusing, please be clear when you are citing a literature results.
7. How is the interplanar spacing calculated in the XRD? It should be mentioned in the experimental section.
8. It was stated that "The result shows that after 2 years of aging, the content of graphitized carbon atoms in graphene oxide increases, and the order of arrangement along its stacking direction is poor." Why is the "order of arrangement" suddenly mentioned here without proper discussion earlier in the XRD results?
9. How is the C/O ratio in XPS determine? Was the relative sensitive factor incorporated in this calculatin?
10. What is the "area" data in Table 1 (XPS)? Is it the absolute area of the deconvoluted peaks? The absolute values of areas of two different samples should not be compared directly. The authors should instead compare the area% of the samples. For example, compare the area% of C=C in XPS C1s for "GO" and "aged GO".
11. In XPS O 1s, the amount of adsorbed water is discussed. However, XPS is operated under vacuum, therefore it is not a good analysis method for adsorbed water content.

12. There are many poorly written sentences in the discussion, for example "The thermal decomposition of graphene oxide is a disproportionation reaction, in which part of the carbon atoms is oxidized and the other part of the carbon atoms is reduced."
13. XPS shows that "aged GO" has adsorbed water, however TGA shows that "aged GO" has more % of adsorbed water?
14. The image quality of Fig5 a and b are very bad.
15. It was stated that "In addition, it is difficult to completely remove the impurity S element during the preparation of graphene oxide. We believe that the weak infrared absorption band near 1168 cm⁻¹ in Figure 6 (a) and Figure 6 (b) should correspond to SO₂" Was the sulfur element detected in XPS?

Author's Response to Decision Letter for (RSOS-202309.R0)

See Appendix A.

RSOS-202309.R1 (Revision)

Review form: Reviewer 2

Is the manuscript scientifically sound in its present form?

Yes

Are the interpretations and conclusions justified by the results?

Yes

Is the language acceptable?

Yes

Do you have any ethical concerns with this paper?

No

Have you any concerns about statistical analyses in this paper?

Yes

Recommendation?

Accept as is

Comments to the Author(s)

The authors have addressed all comments. The manuscript is now technically acceptable for publication. Authors should proofread and check grammar before submitting the final version.

Review form: Reviewer 4

Is the manuscript scientifically sound in its present form?

No

Are the interpretations and conclusions justified by the results?

Yes

Is the language acceptable?

No

Do you have any ethical concerns with this paper?

No

Have you any concerns about statistical analyses in this paper?

Yes

Recommendation?

Major revision is needed (please make suggestions in comments)

Comments to the Author(s)

1. I strongly recommend the authors to use an external language check/proofread service to improve the article's readability. There are too many poorly written sentences, it is difficult for the readers to fully understand the contents.
2. The abstract is very poorly written, it is not up to the standard of an abstract of a research article. Please refer to the guidelines: <https://royalsociety.org/journals/authors/author-guidelines/>
3. Please use consistent acronym after the first introduction. For example, please use GO for graphene oxide after the first introduction of the acronym.
4. There are too many confusing sentences in the Introduction. The content is okay, but the language is not acceptable. Please avoid mentioning too much experimental information in the Introduction section.
5. Is the list of materials in 2.1 section updated? For example, is isopropanol still relevant to this article?
6. Please simplify the Experimental section and avoid stating redundant information. For example the first paragraph in 2.1.
7. Please stick to the acronym that you have introduced in the experimental section. For example, the usage of "GO-a" in the text instead of "graphene oxide aged for 2 years".
8. Is it that GO is 5 days aged, while GO-a is 2 years aged? This information is not clearly stated in the experimental. It is very confusing to differentiate between the samples in the results and discussion section due to the lack of clarity in the experimental section.
9. There is a lack of discussion on the importance of each of the results discussed. Why do the differences between GO and GO-a matter?

10. Are there any technical/instrumental differences that could have affected the changes in the characterization results between GO and GO-a?

Decision letter (RSOS-202309.R1)

Dear Dr Li:

Title: The effect of long-term aging on graphene oxide: structure and thermal decomposition
Manuscript ID: RSOS-202309.R1

The editor assigned to your paper has now received comments from reviewers. We would like you to revise your paper in accordance with the referee and Subject Editor suggestions which can be found below (not including confidential reports to the Editor). Please note this decision does not guarantee eventual acceptance.

Please submit a copy of your revised paper before 09-Sep-2021. Please note that the revision deadline will expire at 00.00am on this date. If we do not hear from you within this time then it will be assumed that the paper has been withdrawn. In exceptional circumstances, extensions may be possible if agreed with the Editorial Office in advance. We do not allow multiple rounds of revision so we urge you to make every effort to fully address all of the comments at this stage. If deemed necessary by the Editors, your manuscript will be sent back to one or more of the original reviewers for assessment. If the original reviewers are not available we may invite new reviewers.

RSC Associate Editor:
Comments to the Author:
(There are no comments.)

RSC Subject Editor:
Comments to the Author:
(There are no comments.)

Reviewers' Comments to Author:

Reviewer: 2

Comments to the Author(s)

The authors have addressed all comments. The manuscript is now technically acceptable for publication. Authors should proofread and check grammar before submitting the final version.

Reviewer: 4

Comments to the Author(s)

1. I strongly recommend the authors to use an external language check/proofread service to improve the article's readability. There are too many poorly written sentences, it is difficult for the readers to fully understand the contents.
2. The abstract is very poorly written, it is not up to the standard of an abstract of a research article. Please refer to the guidelines: <https://royalsociety.org/journals/authors/author-guidelines/>
3. Please use consistent acronym after the first introduction. For example, please use GO for graphene oxide after the first introduction of the acronym.
4. There are too many confusing sentences in the Introduction. The content is okay, but the language is not acceptable. Please avoid mentioning too much experimental information in the Introduction section.
5. Is the list of materials in 2.1 section updated? For example, is isopropanol still relevant to this article?
6. Please simplify the Experimental section and avoid stating redundant information. For example the first paragraph in 2.1.
7. Please stick to the acronym that you have introduced in the experimental section. For example, the usage of "GO-a" in the text instead of "graphene oxide aged for 2 years".

8. Is it that GO is 5 days aged, while GO-a is 2 years aged? This information is not clearly stated in the experimental. It is very confusing to differentiate between the samples in the results and discussion section due to the lack of clarity in the experimental section.

9. There is a lack of discussion on the importance of each of the results discussed. Why do the differences between GO and GO-a matter?

10. Are there any technical/instrumental differences that could have affected the changes in the characterization results between GO and GO-a?

Author's Response to Decision Letter for (RSOS-202309.R1)

See Appendix B.

RSOS-202309.R2

Review form: Reviewer 4

Is the manuscript scientifically sound in its present form?

Yes

Are the interpretations and conclusions justified by the results?

Yes

Is the language acceptable?

Yes

Do you have any ethical concerns with this paper?

No

Have you any concerns about statistical analyses in this paper?

No

Recommendation?

Accept as is

Comments to the Author(s)

The corrections have been done, the paper can be accepted.

Decision letter (RSOS-202309.R2)

Dear Dr Li:

Title: Effect of long-term aging on graphene oxide: structure and thermal decomposition
Manuscript ID: RSOS-202309.R2

It is a pleasure to accept your manuscript in its current form for publication in Royal Society Open Science. The chemistry content of Royal Society Open Science is published in collaboration with the Royal Society of Chemistry.

Yours sincerely,
Dr Ellis Wilde
Publishing Editor, Journals

RSC Associate Editor
Comments to the Author:
(There are no comments.)

RSC Publishing Editor
Comments to the Author:
(There are no comments.)

Reviewer(s)' Comments to Author:
Reviewer: 4

Comments to the Author(s)
The corrections have been done, the paper can be accepted.

Appendix A

List of Response

Dear Editor and Reviewers:

We are grateful for the opportunity to submit a revised version of our manuscript (ID: RSOS-202309. Title: The effect of long-term aging on graphene oxide: structure and thermal decomposition). We appreciate the detailed and constructive comments provided by the reviewers. The manuscript has been carefully revised according to all the suggestions made by the review panel.

To facilitate this discussion, we first retype the comments of the reviewers in italic font and then present our responses to the comments. The revised contents in the manuscript have been highlighted in red, and the contents that can be referred in the revised manuscript are highlighted by bold font in this response letter.

Reply to Reviewer #1

Comments:

1. Please further describe how to control or regulate defects and functional groups and composition during aging.

Response: Thank you very much for this constructive comment. Graphene oxide is placed in a constant temperature location, and the relative humidity is controlled from 20% to 70%.

2. The theoretical analysis should be strengthened, and the change of adsorption energy and binding energy should be further further combined with the thermal analysis experiment or theoretical study in the aging process, so as to support the evolution behavior and related mechanism analysis of defects and functional groups.

Response: Thank you very much for this constructive comment. But I don't have enough time to complete density functional theory simulation calculation. It's a pity for me.

3. According to the experimental results, including the peer experiments, the evolution mechanism of the composition structure in the aging process was further improved.

Response: Thank you very much for this constructive comment. But, the detailed structure of graphene oxide is still unclear. The newly reported model of graphene oxide is dynamic structure model proposed by Dimiev A. M^[1]. Generally, researchers believe that there are epoxy, hydroxyl, carbonyl, and carboxyl groups in graphene oxide, but cannot determine other possible lactones, 1,2-benzoquinone structures, 1,3-epoxy bridges, furans, pyrans, etc. structure. As the aging progresses, CO₂ will be generated and leave a C vacancy on the basal plane in the aged graphene oxide. In addition, oxygen-containing functional groups can also react with moisture and transform into other functional groups. The detailed aging

mechanism is related to the microstructure of graphene oxide. It is not completely certain at this time.

4. *The comparative analysis of similar research work is very necessary, the author please make some supplement.*

Response: Thank you very much for this constructive comment. Kim S et al.^[2] believed that after approximately 35 days, multilayer graphene oxide undergoes spontaneous decomposition and enters a quasi-equilibrium state. After reaching the quasi-equilibrium state, the stability and structure of graphene oxide will change over time^[3]. Chua C K et al.^[3] believe that aging may lead to the reduction of graphene oxide epoxy groups. Storing graphene oxide in an inert atmosphere to avoid light can reduce the decomposition rate of graphene oxide and maintain a high O/C ratio^[3]. In addition, Dimiev A et. al.^[4] believed that the interaction with water would cause changes in graphene oxide.

5. *It is suggested that the highlights and significance of this paper should be further clarified, and the evolution law of defects, functional groups and composition structure during the aging process of GO should be emphasized, and the evolution mechanism should be revealed in combination with the ten revision previews of supporting literature. Related literature, SMALL, 2018, 14(29), 1800987; ADV. MATER., 2014, 26(21): 3484-3489.*

Response: We would like to thank you for this important comment. The instability of graphene oxide may cause the properties of related materials to change during long-term storage/use. This may lead to the failure of key components. This is the purpose of this article.

Reply to Reviewer #2

Comments: *The authors report a detailed study on the changes in thermal decomposition and chemical structure of long-term aged GO under room condition. Particularly, the author studies extensively the thermal decomposition products and thermal stability of the aged GO, both of which provide useful information for academic and industrial users/producers of GO. Therefore, it is suitable for publication in RSOS after addressing the following comments:*

Response: Thank you very much for your encouraging and positive comments on our manuscript. We have listed your concerns below and answered them point-to-point. Here are our responses.

1. *When analyzing the FTIR result on page 4, the author stated that the absorption band at 1620 cm⁻¹ is caused by the aromatic ring vibration. In most GO literature, the scissor bending mode of water can also contribute to the absorption at this wavenumber for GO. When moisture left the GO, this band downshifts to a lower wavenumber, which is what was observed for the 2-year aged GO.*

Response: Thank you for your kindly advice. We agree with this view that the infrared vibration peak near 1622 cm^{-1} is caused by OH bending vibration and aromatic C=C. Aromatic C=C can also produce infrared vibration peaks around 1580 cm^{-1} . Due to the desorption of surface water, the infrared vibration peak caused by OH bending vibration and aromatic C=C near 1622 cm^{-1} is red-shifted to near 1585 cm^{-1} .

2. The authors also observed -CH- stretching for pristine GO in the FTIR spectrum and uses the existence of -CH- stretching to explain the change in layer thickness post aging. The existence of -CH- is quite uncommon in literature. How can the authors be sure that it is not due to contamination? Did the authors collect signals from multiple different samples to confirm this observation?

Response: Thank you for your kindly advice. In FTIR spectrum, the presence of a small amount of -CH- is also observed in other newly prepared graphene oxides. In my opinion, this may be related to the type and quality of raw graphite. The infrared signal of -CH- is very weak, but it does exist. The weak infrared signal caused many researchers to ignore the existence of -CH- and -CH₂-. -CH- and -CH₂- existing in the big hole in the basal plane of graphene oxide has been reported by *Dimiev, A. M. and Eigler S*^[5]. Now, we don't think that the change of -CH- can cause the change of thickness. The change in thickness should be caused by the reduction of oxygen-containing functional groups and the bending of the basal plane. The thickness measured by AFM should be attributed to the two layers of graphene oxide.

3. Page 4, line 30 to 33: "After 2 years of aging, the halfwidth of the diffraction peak of the (001) plane of graphene oxide increased from 0.53527° to 0.91882° , which indicates that the size of the (001) plane of graphene oxide became smaller.". It is better to use interlayer spacing rather than size for more precise expression.

Response: Thank you very much for your constructive comment. The Debye-Scherrer formula was used to calculate the lattice size of graphene oxide. After 2 years of aging, the lattice size of GO reduced from 14.48 nm to 7.11 nm. The FWHM of peaks near 20° has been given in Fig.1 (a). After 2 years of aging, the lattice size of partly reduced GO reduced from 12.16 nm to 1.47 nm.

Debye-Scherrer formula^[6]:

$$L_a = \lambda 0.9\beta^{-1} (\cos \theta)^{-1} \quad (1)$$

In the formula: L_a is the average grain size perpendicular to the crystal plane, λ is the wavelength of X-ray, β is the FWHM of the diffraction peak, θ is the diffraction angle.

The Bragg's equation was used to calculate the interplanar spacing of graphene oxide. After 2 years of aging, the interplanar spacing of high oxidation graphene oxide decreased from 9.00 \AA to 7.59 \AA . After 2 years of aging, the interplanar spacing of partially deoxygenated graphene oxide decreased from 4.51 \AA

to 4.23 Å

Bragg's equation ^[7]:

$$2d \sin \theta = n \lambda \quad (2)$$

In the equation: **d** is the interplanar spacing, **θ** is the incident angle, **n** is the diffraction order, and **λ** is the wavelength of X-ray.

4. On Page 4, when reporting Raman results and calculating D/G ratio of the two GOs, the author concludes that the defect density of sp² region of GO increased based on an increasing D/G ratio from 0.85 to 1.03 of GO after 2 years of aging. Were there enough measurements carry out to support the increase in D/G ratio and is such increase significantly different statistically? The authors should also consider the correlation of D/G ratio vs. the mean inter-defect distance (LD) in the full LD range. (Canc -ado, L. G.; Jorio, A.; Ferreira, E. H. M.; Stavale, F.; Achete, C. A.; Capaz, R. B.; Moutinho, M. V. O.; Lombardo, A.; Kulmala, T.; Ferrari, A. C. *Quantifying Defects in Gra- phene via Raman Spectroscopy at Different Excitation Energies. Nano Lett. 2011, 11, 3190–3196.*) When LD drops below 3 nm, the D/G ratio starts to increase, and this could be applied to GO where the graphene basal plane is highly functionalized with different oxygen groups. In short, it is important to understand that an increase in D/G ratio is not always correlate with a decrease in LD.

Response: Thanks for your important comment. The I_D/I_G is obtained from the average of two sets of newly prepared samples and two sets of aged samples. I resubtracted the baseline of the sample and got a new I_D/I_G value. I_D/I_G of newly prepared GO is 0.87, while the value GO aged 2 years is 0.92. The increase in I_D/I_G does not mean the increase in defects. In this paper, the FWHM of D band of newly prepared GO is 93.98 cm⁻¹, the FWHM of D band of aged GO is 99.19 cm⁻¹. This results show that the L_D of the newly prepared GO and the aged GO are both less than 4 nm. At this time, the larger the I_D/I_G, the lower the defect density of graphene oxide^[8, 9]. *Dimiev, A. M. and Eigler S^[5]* classified the defects in graphene oxide, including S-W defects, oxygen-containing functional group defects, vacancy defects and hole defects. During the aging of graphene oxide, the formation of CO₂ will form vacancy defects on the basal plane of graphene oxide, but the reduction in the number of oxygen-containing functional groups is much greater than the formation of vacancy defects, which makes the average defect distance (L_D) of the aged GO increase, that is, the defect density decreases.

5. On Page 5, the authors report XPS area calculation and areal ratio between pristine and aged GO for the deconvolution of C1s. This is not recommended as XPS area can vary from point to point even for the same sample. Instead, it is recommended that the authors compare the relative composition of each deconvoluted component between the two GOs. To be able to make such comparison, the authors

need to deconvolute at least 3 high resolution C1s spectra for each GO and report standard deviations for each component.

Response: Thanks for your important comment. I have performed multiple fittings to the C1s and O1s high-resolution spectra of GO, and the peak area values of each fitting are very close. But I just returned to the laboratory from the ship 5 days ago, and the deadline for modification is very urgent. I hope I can have time to fit the C1s spectrum more than 3 times to get more credible results. It's a pity for me.

6. Fig 4 can be better presented for better and more effective data visualization. The authors can consider plotting the TGA and MS data (combine Fig 4a and c) in one graph for better presentation of data. The authors can have two separate TGA-MS plots for the two different GOs. Please check Eigler, Siegfried, et al. "Formation and decomposition of CO₂ intercalated graphene oxide." Chemistry of Materials 24.7 (2012): 1276-1282. for reference. It will be easier for readers to understand which gas product is generated at different stages of the thermal decomposition if the data is presented in this manner.

Response: We would like to thank you for your constructive comment, but we are worried that so many mass spectrometry curves make the TG-MS graph complicated for the the lower signal-to-noise ratio of MS spectra.

7. The data set of the normalized ion current intensity vs. temperature curve of O radical ($m/z = 16$) is too noisy for any meaningful interpretation. Did the authors record data for water ($m/z = 18$)? Does the data of water give any useful information?

Response: We would like to thank you for your constructive comment, *the data for water ($m/z = 18$)* has been added to Fig.4 (c). *The data for water ($m/z = 18$) is similar to the data for OH ($m/z = 17$).*

Fig. 4 (c) mass spectrometry of gas products produced by GO-a ($m/z=16, 17, 18, 32$ and 44).

8. *The data visualization of Fig 5 is poor. It is very difficult to read the X-Y axis and I would recommend plotting the data on a contour plot for better presentation.*

Response: Thank you for your advise. **Fig.5** has already been redrawn in the form of contour plots.

Fig.5 Contour plots of infrared absorption spectrum of gaseous products (a) GO, (b) GO-a

9. *Page 6, line 53 to 57: It is not clear what the authors meant by "weak and continuous signal" means and also the significance of a continuous and uninterrupted process. Please revise and clarify the above sentences.*

Response: Thank you for your advise. The sentence has been revised. The 'weak and continuous signal' refers to the infrared signal of gaseous products in the decomposition process of graphene oxide, while the sentence 'a continuous and uninterrupted process' refers to the decomposition process of graphene

oxide.

10. The authors did extensive curve fitting of the fingerprint regime of the FTIR spectra with 17 components. As it stands, this looks overinterpreted. There is no agreed physical model for fitting for the fingerprint regime of GO in the literature and thus the fitting is highly subjective and could easily lead to misinterpretation of the data. If the authors would like to keep this part of the study, I suggest putting them in the supporting information and clearly explain the steps for baseline subtraction and peak fitting.

Response: Thank you for your advise. We agreed with this view the peak fitting in the area will lead to misunderstanding. Therefore, the content of infrared peakfit for GO, GO-a, GO-H2O2 and GO-IPA has been deleted from the text.

11. I suggest author also run DSC measurements of the different GOs and include this data in the study. The DSC measurement of aged GO will give useful information about the change in thermochemistry and kinetics of graphite oxide exothermic decomposition over long term aging, which is important for safety in large-scale storage and processing of GO. Please see Qiu, Yang, et al. "Thermochemistry and kinetics of graphite oxide exothermic decomposition for safety in large-scale storage and processing." Carbon 96 (2016): 20-28 for more details.

Response: We would like to thank you for your constructive comment. The DSC curves of GO (newly prepared) and GO-a (aged 2 years) has been drawn in Fig.8 (a) and Fig.9 (a). Fitting curve of $\ln(\beta/T^2)$ to $1000/T$ has been drawn in Fig.8 (b) and Fig.9 (b), respectively. Calculated by Kissinger formula, the apparent activation energy (E_a) is 150 kJ/mol for GO. After 2 years aging, the apparent activation energy (E_a) decreased to 134 kJ/mol.

Fig.8 (a) DSC curves of GO, (b) Fitting curve of $\ln(\beta/T^2)$ to $1000/T$.

Fig.9 (a) DSC curves of GO-a, (b) Fitting curve of $\ln(\beta/T^2)$ to $1000/T$.

12. In general, the graphics of the figures are poor, please improve the quality.

Response: Thank you for your advise. All the figures has been redrawn.

Reply to Reviewer #3

Comments: The authors report experimental results regarding long-term ageing and thermal decomposition of graphene oxide.

Due to the following concerns the manuscript cannot be accepted for publication in its present form.

The authors must present in more detail the state of the art related to ageing and decomposition of graphene oxide. Graphene oxide after synthesis is known to reach a quasi-equilibrium state with a constant C/O ratio and a stable structure. Reduction mechanisms at room temperature were reported to be highly inefficient, decomposition processes being endothermic. This manuscript report results which stand in contradiction which these features, stating that reduction of graphene oxide takes place during two years of storage at ambient temperature and 70% humidity. More detailed investigations are necessary to prove these conclusions. The presented results are limited to only two samples, one recently synthesized and another sample which was stored for two years. A much more accurate study would contain information related to samples synthesized simultaneously and their structures and composition investigated at regular time intervals.

Response: We would like to sincerely thank you for your advise. Zhou, S. and et. al.^[2, 10] has reported that *graphene oxide after synthesis is known to reach a quasi-equilibrium state with a constant C/O ratio and a stable structure*. However, water will affect the structure and properties of graphene oxide during storage, as described by the dynamic model of graphene oxide proposed by Dimiev Ayrat M and et.al^[1]. The graphene oxide aged 2 years and the newly prepared graphene oxide sample were synthesized in the same preparation process. The newly prepared samples were characterized after about 5 days of

synthesis, and the samples aged 2 years were characterized after the storage time reached 2 years. These two sets of samples are not prepared by two synthetic processes. Now, all the samples prepared at that time have been stored for more than 2 years. Unfortunately, it is not possible to supplement the sample test data stored for several months after preparation.

Reply to Reviewer #4

Comments: Overall, the introduction is clear, but the result and discussion is too long and sometimes difficult to be comprehend. I recommend the authors to simplify the discussions and revise the structures of the sentences. The authors can also consider rearranging some of the minor data into supporting information. Besides that, the authors can use symbols for the samples. For example, "long-term aged GO" and "newly prepared graphene oxide" are too tedious to read.

Is the GO sample before aging characterized 2 years ago? Or a newly synthesized GO sample was characterized to be compared with the aged samples?

Response: We would like to sincerely thank you for your comments on this work. We have thoroughly revised the expressions and rewritten the sentences of the manuscript to improve the quality of our paper. In order to simplify the name, graphene oxide aged for 5 days is represented by ‘GO’, and graphene oxide aged for 2 years is represented by ‘GO-a’. The newly synthesized GO sample characterized 2 years ago is so-called ‘GO after 5 days aging’. The reason we call the newly prepared GO this way is because it takes almost 5 days from the completion of the preparation to the sample test.

1. Can the authors provide a more accurate humidity range for example "xx to 70RH%", rather than just mentioning "not exceed 70%". This is because the humidity is very important for experimental repeatability.

Response: Thank you for your advise. The ‘humidity range’ is from 20% to 70% in this paper (in section 2.3).

2. Is the "humidity box" a commercial dry cabinet? Or is it a self-built device? It is important to provide a clear experimental description.

Response: The "humidity box" is MS-03M dehumidifier bought from MS SHIMEI Electric. We have added temperature and humidity sensors and control programs to the dehumidifier.

3. In Section 2.3, please briefly discuss the purpose of making the GO samples stirred in IPA and H2O2. Again, it is better to denote the samples with symbols.

Response: Thank you for your advise. Other reviewers pointed out that *‘there is no agreed physical model for fitting for the fingerprint regime of GO in the literature and thus the fitting is highly subjective and could easily lead to misinterpretation of the data’*. We agreed with this view very much. Therefore, the content of infrared peakfit for GO, GO-a, GO-H2O2 and GO-IPA will be deleted from the text.

4. In figure 1, which samples are compared? Is it a fresh GO or a 5 days aged GO? It is different with the other analysis?

Response: The newly synthesized GO sample characterized 2 years ago is so-called ‘GO after 5 days aging’. The reason we call the newly prepared GO this way is because it takes almost 5 days from the completion of the preparation to the sample test.

5. In the XRD analysis, the authors stated that “After 2 years of aging, the halfwidth of the diffraction peak of the (001) plane of graphene oxide increased from 0.53527° to 0.91882°, which indicates that the size of the (001) plane of graphene oxide became smaller.” What is "halfwidth"? Is it referring to the FWHM? Please use accurate term in the discussion. Moreover, how do the authors deduce that the size of GO became smaller? Is it proven by the results or referred to another study?

Response: Thank you for your advise. I should use FWHM instead of halfwidth in this paper. We use the Debye-Scherrer formula to calculate the lattice size of GO. After 2 years of aging, the lattice size of GO reduced from 14.48 nm to 7.11 nm. The FWHM of peaks near 20° has been given in **Fig.1 (a)**. After 2 years of aging, the lattice size of partly oxygen-desorption GO reduced from 12.16 nm to 1.47 nm.

Debye-Scherrer formula ^[6]:

$$L_a = \lambda 0.9\beta^{-1} (\cos \theta)^{-1} \quad (1)$$

In the formula: **L_a** is the average grain size perpendicular to the crystal plane, **λ** is the wavelength of X-ray, **β** is the FWHM of the diffraction peak, **θ** is the diffraction angle.

Fig. 1 (a) XRD curves of GO (black) and aged GO-a (red).

6. In the XRD results, it was stated that "The appearance of a broad peak indicates that there are some graphene flakes with lower oxidation degree in the graphene oxide sample aged for 5 years, which is similar to the XRD test results of the reduced graphite oxide at 200°C". This sentence is very confusing, please be clear when you are citing a literature results.

Response: Thank you for your advise. The sentence has been rewritten as follows: **‘The increase in peak intensity near 21.04° indicates that the amount of graphene oxide with a lower degree of oxidation has increased. The graphene oxide heated at 200°C also has a strong peak around 20°.’**

7. How is the interplanar spacing calculated in the XRD? It should be mentioned in the experimental section.

Response: Thank you for your advise. The ‘interplanar spacing’ was calculated by Bragg’s equation.

Bragg’s equation ^[7]:

$$2d \sin \theta = n \lambda \quad (2)$$

In the equation: d is the interplanar spacing, θ is the incident angle, n is the diffraction order, and λ is the wavelength of X-ray.

8. It was stated that "The result shows that after 2 years of aging, the content of graphitized carbon atoms in graphene oxide increases, and the order of arrangement along its stacking direction is poor." Why is the "order of arrangement" suddenly mentioned here without proper discussion earlier in the XRD results?

Response: Thanks for your help. I am sorry for citing their sentence without a good understanding of its meaning. The discussion about *the "order of arrangement"* in this paper is inappropriate. These sentences will be deleted.

9. How is the C/O ratio in XPS determine? Was the relative sensitive factor incorporated in this calculatin?

Response: The C/O ratio was provided by Tsinghua University after considering the relative sensitivity factor.

10. What is the "area" data in Table 1 (XPS)? Is it the absolute area of the deconvoluted peaks? The absolute values of areas of two different samples should not be compared directly. The authors should instead compare the area% of the samples. For example, compare the area% of C=C in XPS C1s for "GO" and "aged GO".

Response: Thank you for your advise. *The "area" data has been revised as follows:*

Tab.1 Positions and areas of graphene oxide C 1s and O 1s deconvolution peaks

Graphene oxide	Feature	C 1s					O 1s			
		C (sp ²)	C (sp ³)	C-O	C=O	O=C-O	C=O	C-O	C-O-C	H ₂ O
GO	Peak (eV)	284.5	285.3	286.6	287.6	288.8	531.0	532.0	532.9	534.0
	Area (%)	34.64	12.11	42.97	6.48	3.80	7.31	38.30	39.03	15.36
GO-a	Peak (eV)	284.6	285.4	286.9	287.7	288.7	531.3	532.2	533.0	534.1
	Area (%)	58.67	8.29	23.83	3.73	5.48	6.51	38.11	43.34	12.05

11. In XPS O 1s, the amount of adsorbed water is discussed. However, XPS is operated under vacuum, therefore it is not a good analysis method for adsorbed water content.

Response: Thank you for your advise. In our view, due to the large specific surface area of graphene oxide (the specific surface area of graphene oxide measured by the methylene blue method is as high as 1800 m²/g.), the vacuum environment during the XPS test may not be enough to cause the complete desorption of surface adsorbed water.

12. There are many poorly written sentences in the discussion, for example "The thermal decomposition of graphene oxide is a disproportionation reaction, in which part of the carbon atoms is oxidized and the other part of the carbon atoms is reduced."

Response: Thank you for your advise. The poorly written sentences in the discussion in this paper has been rewritten.

13. XPS shows that "aged GO" has adsorbed water, however TGA shows that "aged GO" has more % of adsorbed water?

Response: In Fig.7 (e) and (f), we have proven that when the temperature is above 50°C, graphene oxide will decompose to produce CO₂, leaving holes on the sheet of graphene oxide. These holes may become the escape channels of interlayer water and interlayer CO₂ in graphene oxide. The adsorbed water measured in XPS is only the water adsorbed on the surface of graphene oxide, and does not include interlayer water.

14. The image quality of Fig5 a and b are very bad.

Response: Thank you for your advise. **Fig.5** has already been redrawn in the form of contour plots.

Fig.5 Contour plots of infrared absorption spectrum of gaseous products (a) GO, (b) GO-a

15. It was stated that “In addition, it is difficult to completely remove the impurity S element during the preparation of graphene oxide. We believe that the weak infrared absorption band near 1168 cm^{-1} in Figure 6 (a) and Figure 6 (b) should correspond to SO_2 ” Was the sulfur element detected in XPS?

Response: Thank you for your advise. We redrew **Figure 3** which showed the magnification graph of the area inside the box in **Figure 3 (a)**. In **Figure 3 (b)**, the peak near 231 and 168 eV were corresponding to S 2s and S 2p, respectively. The content of

sulfur in graphene oxide is very low, but it still exists after aging.

Fig. 3 (a) Survey XPS spectra, (b) magnification graph of the area inside the box in (a), (c) detailed C 1s and (d) O 1s XPS spectra of GO.

We tried our best to improve the manuscript according to the reviewers' suggestion. We appreciate for editors and reviewers' warm work earnestly, and hope that the correction will meet with approval.

If there are any problems or further processing of the manuscript, please contact us at 15511475485@163.com.

Once again, thank you very much for your comments and suggestions.

Best regards.

Sincerely,

Chen Li

- [1] Dimiev A M, Alemany L B, Tour J M. Graphene oxide. Origin of acidity, its instability in water, and a new dynamic structural model[J]. ACS Nano, 2013, 7(1): 576-588.
- [2] Kim S, Zhou S, Hu Y, et al. Room temperature metastability of multilayer graphene oxide films[J]. Nature Materials, 2012, 11(6): 544-549.
- [3] Chua C K, Pumera M. Light and atmosphere affect the Quasi-equilibrium states of graphite oxide and graphene oxide powders[J]. Small, 2015, 11(11): 1266-1272.
- [4] Dimiev A, Kosynkin D V, Alemany L B, et al. Pristine graphite oxide[J]. Journal of American Chemical Society, 2012, 134(5): 2815-2822.
- [5] Dimiev A M, Eigler S. Graphene oxide: Fundamentals and Applications[M]. India: John Wiley & Sons Inc., 2017.
- [6] Guerrero-Contreras J, Caballero-Briones F. Graphene oxide powders with different oxidation degree, prepared by synthesis variations of the Hummers method[J]. Materials Chemistry and Physics, 2015, 153: 209-220.
- [7] Zaaba N I, Foo K L, Hashim U, et al. Synthesis of graphene oxide using modified Hummers method: solvent influence[J]. Procedia Engineering, 2017, 184: 469-477.
- [8] Lucchese M M, Stavale F, Ferreira E H M, et al. Quantifying ion-induced defects and Raman relaxation length in graphene[J]. Carbon, 2010, 48(5): 1592-1597.
- [9] Cancado L G, Jorio A, Ferreira E H, et al. Quantifying defects in graphene via Raman spectroscopy at different excitation energies[J]. Nano Letters, 2011, 11(8): 3190-3196.
- [10] Zhou S, Bongiorno A. Origin of the chemical and kinetic stability of graphene oxide[J]. Scientific Reports, 2013, 3: 2484-2490.

Appendix B

List of Response

Dear Editor and Reviewers:

We are grateful for the opportunity to submit a revised version of our manuscript (ID: RSOS-202309. Title: Effect of long-term aging on graphene oxide: structure and thermal decomposition). We appreciate the detailed and constructive comments provided by the reviewers. In order to improve the article's readability, I have used an external language check/proofread service.

To facilitate this discussion, we first retype the comments of the reviewers in italic font and then present our responses to the comments. The contents that can be referred in the revised manuscript are highlighted by bold font in this response letter.

Reply to Reviewer #2

Comments:

1. The authors have addresssd all comments. The manuscript is now technically acceptable for publication. Authors should proofread and check grammar before submitting the final version.

Response: Thank you very much for your approval. An external language check/proofread service has been used to improve the articles's readability.

Reply to Reviewer #4

1. I strongly recommend the authors to use an external language check/proofread service to improve the article's readability. There are too many poorly written sentences, it is difficult for the readers to fully understand the contents.

Response: Thank you very much for your advise. An external language check/proofread service has been used to improve the articles's readability.

2. The abstract is very poorly written, it is not up to the standard of an abstract of a

*research article. Please refer to the guidelines:
<https://royalsociety.org/journals/authors/author-guidelines/>.*

Response: Thank you very much for your constructive advise. I have already reread the guidelines: <https://royalsociety.org/journals/authors/author-guidelines/>. The abstract has been rewritten.

3. Please use consistent acronym after the first introduction. For example, please use GO for graphene oxide after the first introduction of the acronym.

Response: In order to avoid misunderstandings, in this paper, we no longer use ‘GO’ to represent graphene oxide. Instead, we use ‘GO’ to represent graphene oxide after 5 days of preparation, and ‘GO-a’ to represent graphene oxide aged 2 years. The use of ‘GO’ to represent generalized graphene oxide in the article has been deleted.

4. There are too many confusing sentences in the Introduction. The content is okay, but the language is not acceptable. Please avoid mentioning too much experimental information in the Introduction section.

Response: Thank you for your advise. Experimental information in the **Introduction section** has been deleted. The language in the **Introduction section** has been revised by an external language check/proofread service.

5. Is the list of materials in 2.1 section updated? For example, is isopropanol still relevant to this article?

Response: I'm sorry that I forgot to delete isopropanol in 2.1 section. Isopropanol is not relevant to this article.

6. Please simplify the Experimental section and avoid stating redundant information. For example the first paragraph in 2.1.

Response: Thank you for your advise. The **Experimental section** has been simplified and redundant information has been deleted.

7. Please stick to the acronym that you have introduced in the experimental section. For

example, the usage of “GO-a” in the text instead of “graphene oxide aged for 2 years”.

Response: Thank you for your advise. We checked and revised the paper according to your suggestions. The “*graphene oxide aged for 2 years*” has been represented by “GO-a”, while the “*graphene oxide prepared after 5 days*” has been represented by “GO”.

8. Is it that GO is 5 days aged, while GO-a is 2 years aged? This information is not clearly stated in the experimental. It is very confusing to differentiate between the samples in the results and discussion section due to the lack of clarity in the experimental section.

Response: Yes, GO is 5 days aged, while GO-a is 2 years aged. I’m sorry that I did not clearly state this information in the paper.

9. There is a lack of discussion on the importance of each of the results discussed. Why do the differences between GO and GO-a matter?

Response: The differences between GO and GO-a reflect the effect of aging on graphene oxide. Changes in the structure and properties of graphene oxide may cause components made of graphene oxide or graphene oxide composite materials as substrates to fail after long-term storage. Related content has been added into the **Conclusion**.

10. Are there any technical/instrumental differences that could have affected the changes in the characterization results between GO and GO-a?

Response: The error of the instrument will affect the test results between **GO** and **GO-a**. Usually, people will call it systematic error. It’s acceptable. In order to avoid the influence of environmental factors, we put the samples to be tested in the same container before testing.

If there are any problems or further processing of the manuscript, please contact us at 15511475485@163.com.

Once again, thank you very much for your comments and suggestions.

Best regards.

Sincerely,

Chen Li